# GENERATIVE FEATURE MATCHING NETWORKS

## ABSTRACT

We propose a non-adversarial feature matching-based approach to train generative models. Our approach, Generative Feature Matching Networks (GFMN), leverages pretrained neural networks such as autoencoders and ConvNet classifiers to perform feature extraction. We perform an extensive number of experiments with different challenging datasets, including ImageNet. Our experimental results demonstrate that, due to the expressiveness of the features from pretrained ImageNet classifiers, even by just matching first order statistics, our approach can achieve state-of-the-art results for challenging benchmarks such as CIFAR10 and STL10.

## 1 INTRODUCTION

One of the key research focus in unsupervised learning is the training of generative methods that can model the observed data distribution. Good progress has been made in recent years with the advent of new approaches such as generative adversarial networks (GANs) (Goodfellow et al., 2014) and variational autoencoders (VAE) (Kingma & Welling, 2013) which use deep neural networks as building blocks. Both methods have advantages and disadvantages, and a significant number of recent works focus on addressing their issues (Radford et al., 2016; Salimans et al., 2016; Kingma et al., 2016; Arjovsky et al., 2017; Chen et al., 2018). While the main disadvantage of VAEs is the generation of blurred images, the main issue with GANs is the training instability due to the adversarial learning.

Feature matching has been explored to improve the stability of GANs (Salimans et al., 2016; Warde-Farley & Bengio, 2017). The key idea in feature matching GANs (FM-GANs) is to use the discriminator network as a feature extractor, and guide the generator to generate data that matches the feature statistics of the real data. Concretely, the objective function of the generator in FM-GAN consists in minimizing the mean squared error of the average features of a minibatch of generated data and a minibatch of real data. The features are extracted from one single layer of the discriminator. FM-GAN is somewhat similar to methods that use maximum mean discrepancy (MMD) (Gretton et al., 2006; 2012). However, while in FM-GAN the objective is to match the mean of the extracted features, in MMD-based generative models (Li et al., 2015; Dziugaite et al., 2015), one normally aims to match all the moments of the two distributions using a Gaussian kernel. Although MMD-based generative models have strong theoretical guarantees, these models normally perform much worse than GANs on challenging benchmarks (Li et al., 2017).

In this work, we focus on answering the following research question: can we train effective generative models by performing feature matching on features extracted from a pretrained neural networks? In other words, we would like to know if adversarial training of the feature extractor together with the generator is a requirement for training effective generators. Towards answering this question, we propose Generative Feature Matching Networks (GFMN), a new feature matching-based approach to train generative models that uses features from pretrained neural networks, breaking away from the problematic min/max game completely. Some interesting properties of the proposed method include: (1) the loss function is directly correlated to the generated image quality; (2) mode collapsing is not an issue; (3) the same pretrained feature extractor can be used across different datasets; and (4) both supervised (classifiers) and unsupervised (autoencoder) models can be used as feature extractors.

We perform an extensive number of experiments with different challenging datasets, including ILSVRC2012 (henceforth Imagenet) (Russakovsky et al., 2015). We demonstrate that, due to the expressiveness of the features from pretrained Imagenet classifiers, even by just matching first order statistics, our approach can achieve state-of-the-art results for challenging benchmarks such as

CIFAR10 and STL10. Moreover, we show that the same feature extractor is effective across different datasets. The main contributions of this work can be summarized as follows: (1) We propose a new effective feature matching-based approach to train generative models that does not use adversarial learning, have stable training and achieves state-of-the-art results; (2) We propose an ADAM-based moving average method that allows effective training with small minibatches; (3) Our extensive quantitative and qualitative experimental results demonstrate that pretrained autoencoders and deep convolutional net (DCNN) classifiers can be effectively used as feature extractors for the purpose of learning generative models.

## 2 GENERATIVE FEATURE MATCHING NETWORKS

### 2.1 THE METHOD

Let $G$ be the generator implemented as a neural network with parameters $\theta$, and let $E$ be a pretrained neural network with $L$ hidden layers. Our proposed approach consists of training $G$ by minimizing the following loss function:

$$\min_{\theta} \sum_{j=1}^{M} ||\mathbb{E}_{x \sim p_{data}} E_j(x) - \mathbb{E}_{z \sim \mathcal{N}(0, I_{n_z})} E_j(G(z; \theta))||^2 \qquad (1)$$

where: $||.||^2$ is the $L_2$ loss; $x$ is a real data point sampled from the data generating distribution $p_{data}$; $z \in \mathbb{R}^{n_z}$ is a noise vector sampled from the normal distribution $\mathcal{N}(0, I_{n_z})$; $E_j(x)$, denotes the output vector/feature map of the hidden layer $j$ from $E$; $M \leq L$ is the number of hidden layers used to perform feature matching.

In practice, we train $G$ by sampling mini-batches of true data and generated (*fake*) data and optimizing the parameters $\theta$ using stochastic gradient descent (SGD) with backpropagation. The network $E$ is used for the purpose of feature extraction only and is kept fixed during the training of $G$.

### 2.2 AUTOENCODER FEATURES

A natural choice of unsupervised method to train a feature extractor is the autoencoder framework. The decoder part of an AE consists exactly of an image generator that uses features extracted by the encoder. Therefore, by design, the encoder network should be a good feature extractor for the purpose of generation.

Let $E$ and $D$ be the encoder and the decoder networks with parameters $\phi$ and $\psi$, respectively. We pretrain the autoencoder using mean squared error (MSE):

$$\min_{\phi, \psi} \mathbb{E}_{p_{data}} ||x - D(E(x; \phi); \psi)||^2$$

or the Laplacian pyramid loss (Ling & Okada, 2006):

$$\mathrm{Lap}_1(x, x') = \sum_j 2^{-2j} |L^j(x) - L^j(x')|_1$$

where $L^j(x)$ is the $j$-th level of the Laplacian pyramid representation of $x$. The Laplacian pyramid loss provides better signal for learning the high frequencies of the images and overcome some of the known issues of the blurry images that one would get with a simple MSE loss. Bojanowski et al. (2018) recently demonstrated that the $\mathrm{Lap}_1$ loss produces better results than $L_2$ loss for both autoencoders and generative models.

Another attractive feature of the autoencoder framework is that the decoder network can be used to initialize the parameters of the generator, which can make the training of the generator easier by starting in a region closer to the data manifold. We use this option in our experiments and show that it leads to significantly better results.

### 2.3 CLASSIFIER FEATURES

Different past work has shown the usefulness and power of the features extracted from DCNNs pretrained on classification tasks (Yosinski et al., 2014). In particular, features from DCNNs pretrained on ImageNet (Russakovsky et al., 2015) have demonstrated an incredible value for a different number

of tasks. In this work, we perform experiments where we use different DCCNs pretrained on ImageNet to play the role of the feature extractor $E$. Our hypothesis is that ImageNet-based features are powerful enough to allow the successful training of (cross-domain) generators by feature matching.

## 2.4 MATCHING FEATURES WITH ADAM MOVING AVERAGE

**From feature matching loss to moving averages.** In order to train with the (mean) feature matching loss, one would need large mini-batches for generating a good estimate of the mean features. When using images larger than $32\times32$ and DCNNs that produce millions of features, this can easily result in memory issues. To alleviate this problem we propose to use moving averages of the difference of means of real and generated data. Instead of computing the (memory) expensive feature matching loss in Eq. 1, we propose to keep moving averages $v_j$ of the difference of feature means at layer $j$ between real and generated data so that

$$||\mathbb{E}_{x\sim p_{data}}E_j(x) - \mathbb{E}_{z\sim\mathcal{N}(0,I_{n_z})}E_j(G(z;\theta))||^2 \approx v_j^\top \left(\frac{1}{N}\sum_{k=1}^N E_j(x_k) - \frac{1}{N}\sum_{k=1}^N E_j(G(z_k;\theta))\right),$$

where $N$ is the minibatch size. Using these moving averages we replace the loss given in Eq. 1 by

$$\min_\theta \sum_{j=1}^M v_j^\top \left(\frac{1}{N}\sum_{k=1}^N E_j(x_k) - \frac{1}{N}\sum_{k=1}^N E_j(G(z_k;\theta))\right), \tag{2}$$

where $v_j$ is a moving average on $\Delta_j$, the difference of the means of the features extracted by the $j$-th layer of $E$:

$$\Delta_j = \frac{1}{N}\sum_{k=1}^N E_j(x_k) - \frac{1}{N}\sum_{k=1}^N E_j(G(z_k;\theta)). \tag{3}$$

The moving average formulation of features matching given in Eq. 2, gives a major advantage on the naive formulation of Eq. 1, since we can now rely on $v_j$ to get a better estimate of the population feature means of real and generated data while using a small minibatch of size $N$. In order to obtain a similar result using the feature matching loss given in Eq. 1, one would need a minibatch with a large size $N$, which becomes problematic as the number of features becomes large.

**ADAM moving average: from SGD to ADAM updates.** Note that for a rate $\alpha$, the moving average $v_j$ has the following update:

$$v_{j,\text{new}} = (1-\alpha) * v_{j,\text{old}} + \alpha * \Delta_j, \forall j = 1\ldots M$$

It is easy to see that the moving average is a gradient descent update on the following loss:

$$\min_{v_j} \frac{1}{2}||v_j - \Delta_j||^2. \tag{4}$$

Hence, writing the gradient update with learning rate $\alpha$ we have equivalently:

$$v_{j,\text{new}} = v_{j,\text{old}} - \alpha * (v_{j,\text{old}} - \Delta_j) = (1-\alpha) * v_{j,\text{old}} + \alpha * \Delta_j.$$

With this interpretation of the moving average we propose to get a better estimate of the moving average using the ADAM optimizer (Kingma & Ba, 2015) on the loss of the moving average given in Eq. 4, such that

$$v_{j,\text{new}} = v_{j,\text{old}} - \alpha ADAM(v_{j,\text{old}} - \Delta_j).$$

$ADAM(x)$ function is computed as follows:

$$m_t = \beta_1 * m_{t-1} + (1-\beta_1) * x$$
$$u_t = \beta_2 * u_{t-1} + (1-\beta_2) * x^2$$
$$\hat{m}_t = m_t/(1-\beta_1^t)$$
$$\hat{u}_t = u_t/(1-\beta_2^t)$$
$$ADAM(x) = \hat{m}_t/(\sqrt{\hat{u}_t} + \epsilon)$$

where $x$ is the gradient for the loss function in Eq. 4, $t$ is the iteration number, $m_t$ is the first moment vector at iteration $t$, $u_t$ is the second moment vector at iteration $t$, $\beta_1 = .9$, $\beta_2 = .999$ and $\epsilon = 10^{-8}$

are constants. $m_0$ and $u_0$ are initialized as proposed by Kingma & Ba (2015). We refer the reader to (Kingma & Ba, 2015) for a more detailed description of the ADAM optimizer.

This moving average formulation, which we call *ADAM Moving Average* (AMA) promotes stable training when using small minibatches. The main advantage of AMA over simple moving average (MA) is in its adaptive first order and second order moments that ensures a stable estimation of the moving averages $v_j$. In fact, this is a non stationary estimation since the mean of the generated data changes in the training, and it is well known that ADAM works well for such online and non stationary losses (Kingma & Ba, 2015).

In Section 4.2.3 we provide experimental results supporting: (1) The memory advantage that the AMA formulation of feature matching offers over the naive implementation of feature matching; (2) The stability advantage and improved generation results that AMA allows when compared to the naive implementation of the simple MA.

## 3 Related work

Features from DCNNs pretrained on ImageNet have been used frequently to perform transfer learning in many computer vision tasks (Huh et al., 2016). Some previous work uses DCNN features in the context of image generation and transformation. Dosovitskiy & Brox (2016) combines feature based loss with adversarial loss to improve image quality of variational autoencoders (VAE) (Kingma & Welling, 2013). Johnson et al. (2016) proposes a feature based loss that uses features from different layers of the VGG-16 neural network and is effective for image transformation task such as style transfer and super-resolution. Johnson et al. (2016) confirms the findings of Mahendran & Vedaldi (2015) that the initial layers of the network are more related to content while the last layers are more related to style.

Our proposed approach is closely related to the recent body of work on MMD-based generative models (Li et al., 2015; Dziugaite et al., 2015; Li et al., 2017; Bikowski et al., 2018; Ravuri et al., 2018). In fact, our method is a type of MMD where we (only) match the first moment of the transformed data. Among the approaches reported in the literature, the closest to our method is the Generative Moment Matching Network + Autoencoder (GMMN+AE) proposed by Li et al. (2015). In GMMN+AE, the objective is to train a generator $G$ that maps from a prior uniform distribution to the latent code learned by a pretrained AE. To generate a new image, one samples a noise vector $z$ from the prior, maps it the AE latent space using $G$, then uses the (frozen) decoder to map from the AE latent space to the image space. One key difference in our approach is that our generator $G$ maps from the $z$ space directly to the data space, such as in GANs (Goodfellow et al., 2014). Additionally the dimensionality of the feature space that we use to perform distribution matching is orders of magnitude larger than the dimensionality of the latent code normally used in GMMN+AE. Li et al. (2017) demonstrate that GMMN+AE is not competitive with GANs for challenging datasets such as CIFAR10. Recent MMD-based generative models have demonstrated state-of-the-art results with the use of adversarial learning to train the MMD kernel as a replacement of the fixed Gaussian kernel in GMMN (Li et al., 2017; Bikowski et al., 2018). Additionally, Ravuri et al. (2018) recently proposed a method to perform online learning of the moments while training the generator. Our proposed method differs from these previous approaches where we use a frozen pretrained feature extractor to perform moment matching.

Bojanowski et al. (2018) proposed the Generative Latent Optimization (GLO) model, a generative approach that jointly optimizes the model parameters and the noise input vectors $z$. GLO models obtain competitive results for CelebA and LSUN datasets without using adversarial training. Bojanowski et al. (2018) also demonstrated that the Laplacian pyramid loss is an effective way to improve the performance of non-adversarial methods that use reconstruction loss. Our work relates also to plug and play generative models of Nguyen et al. (2017) where a pretrained classifier is used to sample new images, using MCMC sampling methods.

Our work is also related to AE-based generative models variational autoencoder (VAE) (Kingma & Welling, 2013), adversarial autoencoder (AAE) (Makhzani et al., 2016) and Wasserstein autoencoder (WAE) (Tolstikhin et al., 2018). While in VAE and WAE a penalty is used to impose a prior distribution on the hidden code vector of the AE, in AAE an adversarial training procedure is used

for that purpose. In our method, the aim is to get a generative model out of a pretrained autoencoder. We fix the pretrained encoder to be the discriminator in a GAN like setting.

Another recent line of work that involves the use of AEs in generative models consists in applying AEs to improve GANs stability. Zhao et al. (2017) proposed an energy based approach where the discriminator is replaced by an autoencoder. Warde-Farley & Bengio (2017) augments the training loss of the GAN generator by including a feature reconstruction loss term that is computed as the mean squared error of a set of features extracted by the discriminator and their reconstructed version. The reconstruction is performed using an AE trained on the features extracted by the discriminator for the real data.

## 4 EXPERIMENTS AND RESULTS

### 4.1 EXPERIMENTAL SETUP

**Datasets:** We evaluate our proposed approach on MNIST (LeCun et al., 1998) (60k training, 10k test images, 10 classes), CIFAR10 (Krizhevsky, 2009) (50k training, 10k test images, 10 classes), STL10 (Coates et al., 2011) (5K training, 8k test images, 100k unlabeled, 10 classes), CelebA (Liu et al., 2015) (200k images) and different portions of ImageNet (Russakovsky et al., 2015) datasets. MNIST and STL10 images are rescaled to 32×32, while CelebA and ImageNet images are rescaled to 64×64. CelebA images are center cropped to 160×160 before rescaling.

**GFMN Generator:** In our experiments with all datasets but ImageNet, our generator $G$ uses a DCGAN-like architecture (Radford et al., 2016). For CIFAR10, STL10 and CelebA, we use two extra layers as commonly used in previous works (Mroueh & Sercu, 2017; Gulrajani et al., 2017). For ImageNet, we use a Resnet-based generator such as the one in (Miyato et al., 2018). More details about the architectures can be found in Appendix A.2.

**Autoencoder Features**: For most AE experiments, we use an encoder network whose architecture is similar to the discriminator in DCGAN (strided convolutions). We use batch normalization and ReLU non-linearity after each convolution. We set the latent code size to 8, 128, 128, and 512 for MNIST, CIFAR10, STL10 and CelebA, respectively. To perform feature extraction, we get the output of each ReLU in the network as well as the output of the very last layer, the latent code. Additionally, we also perform some experiments where the encoder uses a VGG13 architecture. The decoder network $D$ uses a network architecture similar to our generator $G$. More details in Appendix A.2.

**Classifier Features:** We perform our experiments on classifier features with VGG19 (Simonyan & Zisserman, 2014) and Resnet18 networks (He et al., 2016) which we pretrained using the whole ImageNet dataset with 1000 classes. More details about the pretrained ImageNet classifiers can be found in Appendices A.2 and A.3.

**GFMN Training:** We train GFMN with an ADAM optimizer and keep most of the hyperparameters fixed for the different datasets. We use $n_z = 100$ and minibatch size 64. When using autoencoder features, we set the learning rate to $5 \times 10^{-6}$ when $G$ is initialized with $D$, and to $5 \times 10^{-5}$ when it is not. When using features from ImageNet classifiers, we set set the learning rate to $1 \times 10^{-4}$. We use ADAM moving average (Sec. 2.4) in all reported experiments.

### 4.2 EXPERIMENTAL RESULTS

#### 4.2.1 AUTOENCODER FEATURES AND GENERATOR INITIALIZATION

In this section, we present experimental results on the use of pretrained encoders as feature extractors. The first two rows of results in Tab. 1 show GFMN performance in terms of Inception Score (IS) (Salimans et al., 2016) and Fréchet Inception Distance (FID) (Heusel et al., 2017) for CIFAR10 in the case where the (DCGAN-like) encoder is used as feature extractor. The use of a pretrained decoder $D$ to initialize the generator gives a significant boost in both IS and FID. A visual comparison that corroborates the quantitative results can be found in Appendix A.5. In Figures 1a, 1b, and 1c, we present random samples generated by GFMN when trained with MNIST, CIFAR10 and CelebA datasets, respectively. For each dataset, we train its respective AE using the (unlabeled) training set.

Table 1: CIFAR10 results for GFMN with different feature extractors.

| Feature Extractor | Pre-trained On | # features | Initialize $G$ | Inception Score | FID (5K/50K) |
|---|---|---|---|---|---|
| Encoder | CIFAR10 | 60K | × | $3.76 \pm 0.04$ | 96.5 / 92.5 |
| | | | ✓ | $4.43 \pm 0.05$ | 73.9 / 69.6 |
| Encoder (VGG13) | CIFAR10 ImageNet | 244K | ✓ ✓ | $4.60 \pm 0.06$ $4.90 \pm 0.07$ | 60.8 / 56.5 59.9 / 55.5 |
| Resnet18 | ImageNet | 544K | × | $7.03 \pm 0.11$ | 35.7 / 31.1 |
| | | | ✓ | $7.25 \pm 0.07$ | 32.2 / 27.4 |
| VGG19 | ImageNet | 296K | × | $7.42 \pm 0.09$ | 27.5 / 22.8 |
| | | | ✓ | $7.71 \pm 0.06$ | 26.9 / 22.4 |
| VGG19 + Resnet18 | ImageNet | 832K | × | $7.67 \pm 0.08$ | 23.5 / 19.0 |
| | | | ✓ | $\mathbf{7.99 \pm 0.06}$ | **23.1 / 18.5** |

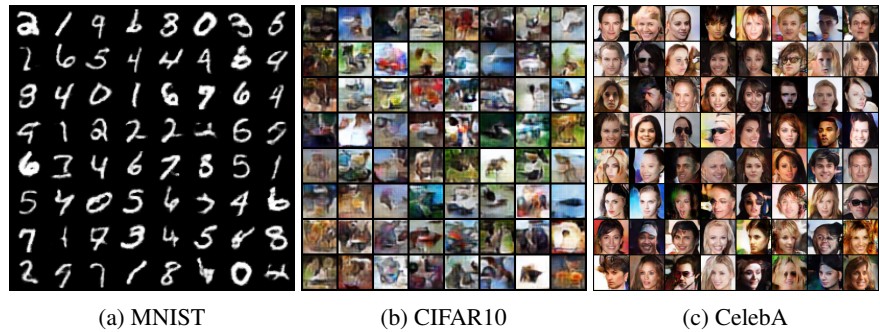

| (a) MNIST | (b) CIFAR10 | (c) CelebA |
|---|---|---|

Figure 1: Generated samples from GFMN using pretrained encoder as feature extractor.

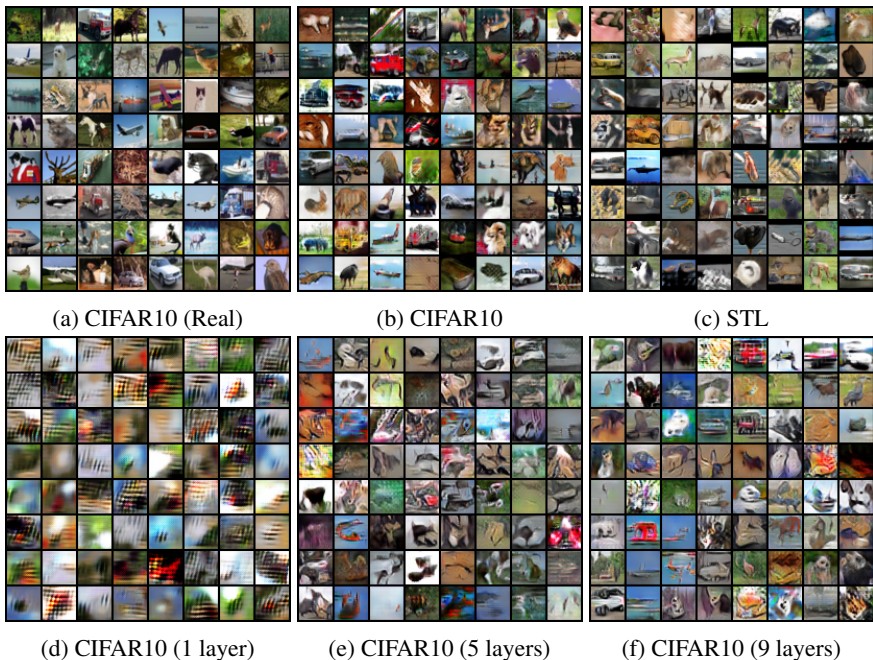

(a) CIFAR10 (Real)  (b) CIFAR10  (c) STL

(d) CIFAR10 (1 layer)  (e) CIFAR10 (5 layers)  (f) CIFAR10 (9 layers)

Figure 2: Generated samples from GFMN that uses as feature extractor a VGG-19 net pretrained on ImageNet. (2a) is a sample from the (real) CIFAR10 dataset. (2d – 2f) show the impact of using different number of layers to perform feature matching.

### 4.2.2 (Cross-domain) Classifier Features

The last six rows in Tab. 1 present the IS and FID for our best configurations that use ImageNet pretrained VGG19 and Resnet18 as feature extractors. Yhere is a large boost in performance when ImageNet classifiers are used as feature extractors instead of autoencoders. Despite the classifiers being trained on data from a different domain (ImageNet vs. CIFAR10), the classifier features are significantly more effective. In all cases, the use of an initialized generator improves the results. However, the improvements are much less significant when compared to the one obtained for the encoder feature extractor. We perform an additional experiment where we use simultaneously VGG19 and Resnet18 as feature extractors, which increases the number of features to 832K. This last configuration gives the best performance for both CIFAR10 and STL10. Figures 2b and 2c show random samples from the GFMN$^{\text{VGG19+Resnet18}}$ model, where no init. of the generator is used.

In Tab. 2, we report IS and FID for increasing number of layers (i.e. number of features) in our extractors VGG19 and Resnet18. We select up to 16 layers for VGG19 and 17 layers for Resnet18, which means that we excluded the output of fully connected layers. Using more layers dramatically improves the performance of both feature extractors, reaching (IS) peak performance when the maximum number of layers is used. The results in Tab. 1 are better than the ones in Tab. 2 because, for the former, we trained the models for a longer number of epochs. All models in Tab. 2 are trained for 391K generator updates, while VGG19 and Resnet18 models in Tab. 1 are trained for 1.17M updates (we use small learning rates). Note that for both feature extractors, the features are ReLU activation outputs. As a result, the encodings may be quite sparse. Figs. 2d, 2e and 2f show generated images when 1, 3, and 9 layers are used for feature matching, respectively (more in Appendix A.7).

In order to check if the number of features is the main factor for the performance, we performed an experiment where we trained an autoencoder whose encoder network uses a VGG13 architecture. This encoder produces a total of 244K features. We pretrained the autoencoder we both CIFAR10 and ImageNet datasets, so to compare the impact of the autoencoder training set size. The results for this experiment are in the 3rd and 4th rows of Tab. 1 (Encoder (VGG13)). Although there is some improvement in both IS and FID, specially when using the Encoder pretrained with ImageNet, the boost is not comparable with the one obtained by using a VGG19 classifier. In other words, features from classifiers are significantly more informative than autoencoder features for the purpose of training generators by feature matching.

Table 2: Impact of the number of layers/features used for feature matching in GFMN (1K=$2^{10}$).

| | VGG19 | | | Resnet18 | | |
|---|---|---|---|---|---|---|
| # layers | # features | IS | FID (5K / 50K) | # features | IS | FID (5K / 50K) |
| 1 | 64K | $3.59 \pm 0.05$ | 176.0 / 172.9 | 64K | $3.47 \pm 0.04$ | 189.3 / 185.4 |
| 3 | 160K | $5.13 \pm 0.04$ | 86.2 / 81.9 | 192K | $3.91 \pm 0.03$ | 102.2 / 98.1 |
| 5 | 208K | $5.94 \pm 0.08$ | 60.4 / 55.9 | 320K | $4.72 \pm 0.05$ | 86.9 / 82.5 |
| 7 | 240K | $6.49 \pm 0.07$ | 46.6 / 42.2 | 384K | $5.27 \pm 0.04$ | 76.6 / 72.1 |
| 9 | 264K | $7.03 \pm 0.07$ | 37.8 / 33.3 | 448K | $5.35 \pm 0.06$ | 65.7 / 61.4 |
| 11 | 280K | $7.37 \pm 0.06$ | 32.5 / 28.0 | 480K | $6.28 \pm 0.07$ | 55.3 / 50.9 |
| 13 | 290K | $7.27 \pm 0.09$ | 31.4 / 26.9 | 512K | $6.25 \pm 0.07$ | 47.3 / 42.4 |
| 15 | 294K | $7.24 \pm 0.07$ | **29.9 / 25.5** | 528K | $6.43 \pm 0.08$ | 43.3 / 38.7 |
| 16/17 | 296K | **$7.44 \pm 0.09$** | 32.3 / 27.6 | 544K | **$6.92 \pm 0.08$** | **35.3 / 30.8** |

### 4.2.3 Adam Moving Average and Training Stability

In this section, we present experimental results that evidence the advantage of our proposed ADAM moving average (AMA) over the simple moving average (MA). The main benefit of AMA is the promotion of stable training when using small minibatches. The ability to train with small minibatches is *essential* due to GFMN's need for large number of features from DCNNs, which becomes a challenge in terms of GPU memory usage. For instance, our Pytorch (Paszke et al., 2017) implementation of GFMN can only handle minibatches of size up to 160 when using VGG19 as a feature extractor and image size 64×64 on a Tesla K40 GPU w/ 12GB of memory. A more optimized implementation minimizing Pytorch's memory overhead could, in principle, handle somewhat larger minibatch sizes (as could a more recent Tesla V100 w/ 16 GB). However, if we increase the image

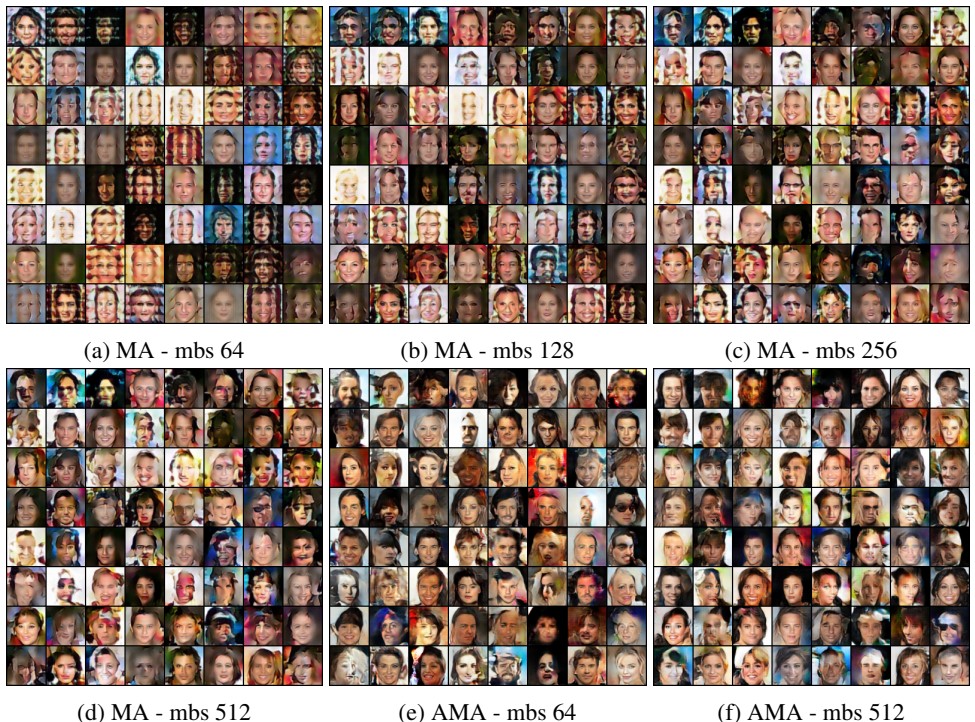

(a) MA - mbs 64        (b) MA - mbs 128        (c) MA - mbs 256

(d) MA - mbs 512        (e) AMA - mbs 64        (f) AMA - mbs 512

Figure 3: Generated images from GFMN trained with either simple Moving Average (MA) (3a, 3b, 3c and 3d) or Adam Moving Average (AMA) (3e and 3f), and different minibatch sizes (mbs). While small minibatch sizes have a big negative effect for MA, it is not an issue for AMA.

size or the feature extractor size, the memory footprint increases quickly and we will always run out of memory when using larger minibatches, regardless of implementation or hardware.

For the experiments presented in this section, we use CelebA as the training set, and the feature extractor is the encoder from an autoencoder that follows a DCGAN-like architecture. We use this feature extractor because it is smaller than VGG19/Resnet18 and allows for minibatches of size up to 512 for image size 64×64. Figure 3 shows generated images from GFMN when trained with either MA or our proposed AMA. For MA, we present generated images for GFMN trained with four different batch sizes: 64, 128, 256 and 512 (Figs. 3a, 3b, 3c and 3d, respectively). For AMA, we show results for two different minibatch sizes: 64 and 512 (Figs. 3e and 3f, respectively). We can note that the minibatch size has a huge impact in the quality of generated images when training with MA. With minibatches smaller than 512 (Figs. 3a, 3b and 3c), almost all images generated by GFMN trained with MA are quite damaged. On the other hand, when using AMA, GFMN generates much better images even with minibatch size 64 (Fig. 3e). For AMA, increasing the minibatch size from 64 to 512 (Fig. 3f) does not seem to improve the quality of generated images for the given dataset and feature extractor. In Appendix A.9, we show a comparison between MA and AMA when VGG19 ImageNet classifier is used as a feature extractor. A minibatch size of 64 is used for that experiment. We can see in Fig. 10 that AMA also has a very positive effect in the quality of generated images when a stronger feature extractor is used. An alternative for training with larger minibatches would be the use of multi-GPU, multi-node setups. However, performing large scale experiments is beyond the scope of the current work. Moreover, many practitioners do not have access to a GPU cluster, and the availability of methods that can work on a single GPU with small memory footprint is essential.

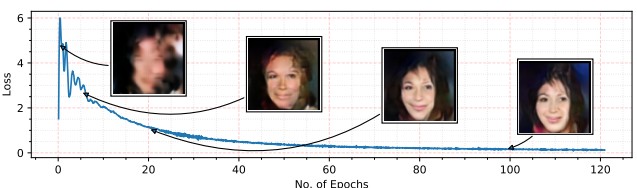

Figure 4: Loss as a function of training epochs with example of generated faces.

An important advantage of GFMN over adversarial methods is its training stability. Fig. 4 shows the evolution of the generator loss per epoch with some generated examples for an experiment where AMA is used. There is a clear correlation between the quality of generated images and the loss. Moreover, mode collapsing was not observed in our experiments with AMA.

### 4.2.4 IMAGENET EXPERIMENTS

In order to evaluate the performance of GFMN for an even more challenging dataset, we trained GFMN[VGG19] with different portions of the ImageNet dataset. Fig. 5a shows some (cherry picked) images generated by GFMN[VGG19] trained on the ImageNet subset that contains different dog breeds (same as used in (Zhao et al., 2017)). The results are quite impressive given that we perform unconditional generation. Fig. 5b presents (randomly sampled) images generated by GFMN[VGG19] trained with the *daisy* portion of ImageNet. More results can be found in Appendix A.1.

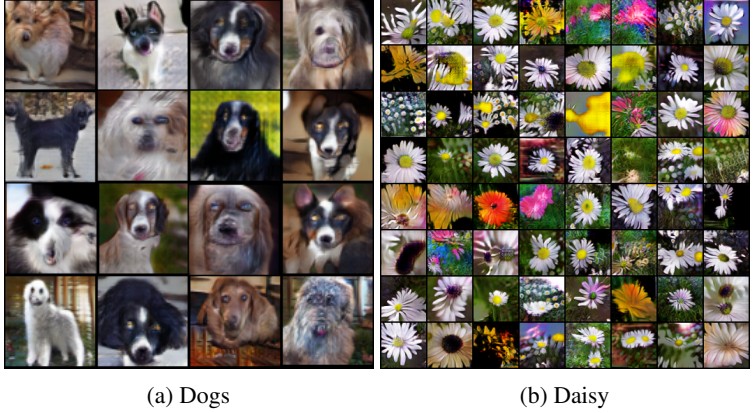

(a) Dogs            (b) Daisy

Figure 5: Generated samples from GFMN trained for two different portions of ImageNet dataset.

### 4.2.5 COMPARISON WITH STATE-OF-THE-ART APPROACHES

In Tab. 3, we compare GFMN results with different adversarial and non-adversarial approaches for CIFAR10 and STL10. The table includes results for recent models that, like ours, use a DCGAN-like (or CNN) architecture in the generator and do not use CIFAR10/STL10 labels while training the generator. Despite using a frozen cross-domain feature extractor, GFMN outperforms the other systems in terms of FID for both datasets, and achieves the best IS for CIFAR10.

We performed additional experiments with a WGAN-GP architecture where: (1) the discriminator is a VGG19 or a Resnet18; (2) the discriminator is pretrained on ImageNet; (3) the generator is pretrained on CIFAR10 through autoencoding. The objective of the experiment is to evaluate if WGAN-GP can benefit from DCNN classifiers pretrained on ImageNet. Although we tried different hyperparameter combinations, we were not able to successfully train WGAN-GP with VGG19 or Resnet18 discriminators. More details about this experiment in Appendix A.8.

### 4.3 DISCUSSION

This work is driven towards answering the question of whether one can train effective generative models by performing feature matching on features extracted from pretrained neural networks. The goal is to avoid adversarial training, breaking away from the problematic min/max game completely. According to our experimental results, the answer to our research question is *yes*. We achieve successful non-adversarial training of generative feature matching networks by introducing different key ingredients: (1) a more robust way to compute the moving average of the mean features by using ADAM optimizer, which allows us to use small minibatches; (2) the use of features from all layers of pretrained neural networks; (3) the use of features from multiple neural networks at the same time (VGG19 + Resnet18); and (4) the initialization of the generator network.

Our quantitative results in Tab. 3 show that GFMN achieves better or similar results compared to the state-of-the-art Spectral GAN (SN-GAN) (Miyato et al., 2018) for both CIFAR10 and STL10. This is an impressive result for a non-adversarial feature matching-based approach that uses pretrained cross-domain feature extractors and has stable training. When compared to other MMD approaches

Table 3: Inception Score and FID of different generative models for CIFAR10 and STL10.

| Model | CIFAR 10 | | STL 10 | |
|---|---|---|---|---|
| | IS | FID (5K / 50K) | IS | FID (5K / 50K) |
| Real data | 11.24±.12 | 7.8 / 3.2 | 26.08±.26 | 8.08 / 4.0 |
| **No Adversarial Training** | | | | |
| GMMN (Li et al., 2017) | 3.47±.03 | | | |
| GMMN+AE (Li et al., 2017) | 3.94±.04 | | | |
| VAE (Lucic et al., 2017) | - | 155.7 / - | | |
| (ours) GFMN$^{\text{VGG+Resnet}}$ | 7.67 ± 0.08 | 23.5 / 19.0 | 8.45 ± 0.09 | 36.2 / 18.8 |
| (ours) GFMN$^{\text{VGG+Resnet}}_{\text{Ginit}}$ | **7.99 ± 0.06** | **23.1 / 18.5** | 8.23 ± 0.13 | **34.8 / 18.1** |
| **Adversarial Training & Online Moment Learning Methods** | | | | |
| ALI (Dumoulin et al., 2017) | 5.34±.05 | | | |
| BEGAN (Berthelot et al., 2017) | 5.62 | | | |
| MMD GAN (Li et al., 2017) | 6.17±.07 | | | |
| MMD$_{dist}$ GAN (Bikowski et al., 2018) | 6.39±.04 | 40.2 / - | | |
| WGAN (Miyato et al., 2018) | 6.41±.11 | 42.6 / - | 7.57±.10 | 64.2 |
| MMD$_{rq}$ GAN (Bikowski et al., 2018) | 6.51±.03 | 39.9 / - | | |
| WGAN-GP (Miyato et al., 2018) | 6.68±.06 | 40.2 / - | 8.42±.13 | 55.1 / - |
| McGAN (Mroueh et al., 2017) | 6.97±.10 | | | |
| SN-GANs (Miyato et al., 2018) | 7.58±.12 | 25.5 / - | **8.79±.14** | 43.2 / - |
| MoLM-1024 (Ravuri et al., 2018) | 7.55±.08 | 25.0 / 20.3 | | |
| MoLM-1536 (Ravuri et al., 2018) | 7.90±.10 | 23.3 / 18.9 | | |

(Li et al., 2015; Dziugaite et al., 2015; Li et al., 2017; Bikowski et al., 2018; Ravuri et al., 2018), GFMN presents important distinctions (some of them already listed in Sec. 3) which make it an attractive alternative. Compared to GMMN and GMMN+AE (Li et al., 2015), we can see in Table 3 that GFMN achieves far better results. In Appendix A.10, we also show a qualitative comparison between GFMN and GMMN results. The main reason why GFMN results are significantly better than GMMN is because GFMN uses a strong, robust kernel function (a pretrained DCNN), which, together with our AMA trick, allows the training with small minibatches. On the other hand, the Gaussian kernel used in GMMN requires a very large minibatch size in order to work well, which is impractical due to memory limitations and computational cost. Compared to recent adversarial MMD methods (MMD GAN) (Li et al., 2017; Bikowski et al., 2018) GFMN also presents significantly better results while avoiding the problematic min/max game. GFMN achieves similar results to the Method of Learned Moments (MoLM) (Ravuri et al., 2018), while using a much smaller number of features to perform matching. The best performing model from Ravuri et al. (2018), MoLM-1536, uses around 42 million moments to train the CIFAR10 generator, while our best GFMN model uses around 850 thousand moments/features only, almost 50x less. In other words, MoLM-1536 can be used in large-scale environments only, while GFMN can be used in single GPU environments.

One may argue that the best results from GFMN are obtained with feature extractors that were trained in a supervised manner (classifiers). However, there are two important points to note: (1) we use a cross domain feature extractor and do not use labels from the target datasets (CIFAR10, STL10, MNIST, CelebA); (2) since the accuracy of the classifier does not seem to be the most important factor for generating good features (VGG19 classifier produces better features although it is less accurate than Resnet18, see Appendix A.3); we are confident that GFMN will also achieve state-of-the-art results when trained with features from classifiers trained using unsupervised methods such as the one recently proposed by Caron et al. (2018).

## 5 CONCLUSION

In this work, we introduced GFMN, an effective non-adversarial approach to train generative models. GFMNs are demonstrated to achieve state-of-the-art results while avoiding the challenge of defining and training an adversarial discriminator. Our feature extractors can be easily obtained and provide for a robust and stable training of our generators. Some interesting open questions include: what type of feature extractors other than classifiers and auto-encoders are good for GFMN? What architecture designs are better suited for the purpose of feature extraction in GFMN?

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

# A  APPENDIX

## A.1  GFMN APPLIED TO IMAGENET

We trained GFMN$^{\text{VGG19}}$ with different portions of the ImageNet dataset using images of size 64×64. Although we adopted this image size due to speedup and memory efficient purposes, GFMN is also effective to generate images of larger sizes. Fig. 6 shows (randomly sampled) images generated by GFMN$^{\text{VGG19}}$ trained with the following ImageNet portions: pizza (Fig. 6a), daisy (Fig. 6b), breeds of dogs (Fig. 6c) and Persian cats (Fig. 6d). Note that in this experiment the generators are trained from scratch, there is not initialization of the generators. While *pizza*, *daisy* and *Persian cats* consist in single ImageNet classes, the *breeds of dogs* portion (same as use in (Zhao et al., 2017)) consists in multiple classes and therefore is a more challenging task since we doing unconditional generation. For the experiments with ImageNet, we use a Resnet generator similar to the one used by Miyato et al. (2018).

## A.2  NEURAL NETWORK ARCHITECTURES

In Tables 4 and 5 we detail the neural net architectures used in our experiments. In both DCGAN-like generator and discriminator, an extra layer is added when using images of size 64×64. In VGG19 architecture, after each convolution, we apply batch normalization and ReLU. The Resnet generator is used for ImageNet experiments only.

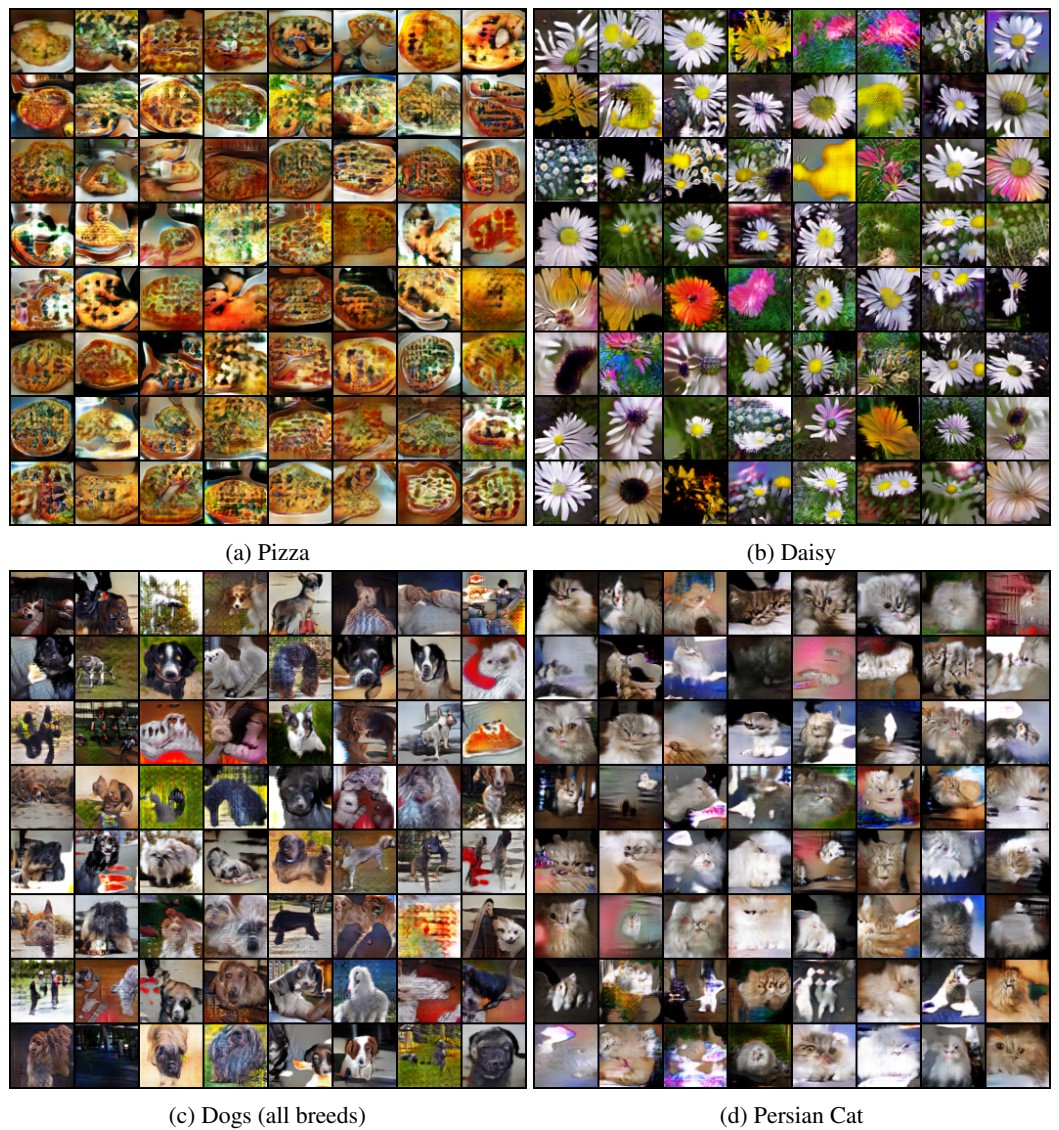

(a) Pizza

(b) Daisy

(c) Dogs (all breeds)

(d) Persian Cat

Figure 6: Random samples from GFMN models using different portions of ImageNet as training set. A VGG19 pretrained on ImageNet is use as feature extractor. Images are generated in size 64x64.

| $z \in \mathbb{R}^{100} \sim \mathcal{N}(0, I)$ |
|---|
| dense $\to 4 \times 4 \times 512$ |
| $4 \times 4$, stride=2 Deconv BN 256 ReLU |
| $4 \times 4$, stride=2 Deconv BN 128 ReLU |
| $4 \times 4$, stride=2 Deconv BN 64 ReLU |
| $3 \times 3$, stride=1 Conv 3 BN 64 ReLU |
| $3 \times 3$, stride=1 Conv 3 BN 64 ReLU |
| $3 \times 3$, stride=1 Conv 3 Tanh |

DCGAN like Generator

ResBlock

| $z \in \mathbb{R}^{100} \sim \mathcal{N}(0, I)$ |
|---|
| dense, $4 \times 4 \times 1024$ |
| ResBlock up 512 |
| ResBlock up 256 |
| ResBlock up 128 |
| ResBlock up 164 |
| BN, ReLU, $3 \times 3$ conv 3 |
| Tanh |

Resnet Generator

Table 4: Generators

| RGB image $x \in \mathbb{R}^{m \times m \times 3}$ | |
|:---:|:---:|
| $\begin{matrix} 3 \times 3, & 64 \\ 3 \times 3, & 64 \end{matrix}$ | $\times 1$ |
| maxpool | |
| $\begin{matrix} 3 \times 3, & 128 \\ 3 \times 3, & 128 \end{matrix}$ | $\times 1$ |
| maxpool | |
| $\begin{matrix} 3 \times 3, & 256 \\ 3 \times 3, & 256 \end{matrix}$ | $\times 2$ |
| maxpool | |
| $\begin{matrix} 3 \times 3, & 512 \\ 3 \times 3, & 512 \end{matrix}$ | $\times 2$ |
| maxpool | |
| $\begin{matrix} 3 \times 3, & 512 \\ 3 \times 3, & 512 \end{matrix}$ | $\times 2$ |
| maxpool | |
| FC $\rightarrow$ 1000, softmax | |

VGG19

| RGB image $x \in \mathbb{R}^{m \times m \times 3}$ |
|:---:|
| $4 \times 4$,stride=2 Conv 64 ReLU |
| $3 \times 3$,stride=1 Conv 64 ReLU |
| $3 \times 3$,stride=1 Conv 64 ReLU |
| $4 \times 4$,stride=2 Conv 128 ReLU |
| $4 \times 4$,stride=2 Conv 256 ReLU |
| dense (FC) $\rightarrow$ 100 |

DCGAN like Encoder

| RGB image $x \in \mathbb{R}^{m \times m \times 3}$ | |
|:---:|:---:|
| $7 \times 7$ conv, 64, stride 2 | |
| $3 \times 3$ max pool, stride 2 | |
| $\begin{matrix} 3 \times 3, & 64 \\ 3 \times 3, & 64 \end{matrix}$ | $\times 2$ |
| $\begin{matrix} 3 \times 3, & 128 \\ 3 \times 3, & 128 \end{matrix}$ | $\times 2$ |
| $\begin{matrix} 3 \times 3, & 256 \\ 3 \times 3, & 256 \end{matrix}$ | $\times 2$ |
| $\begin{matrix} 3 \times 3, & 512 \\ 3 \times 3, & 512 \end{matrix}$ | $\times 2$ |
| average pool | |
| FC $\rightarrow$ 1000, softmax | |

Resnet18

Table 5: Feature extractor architectures. $m = 32$ for MNIST, CIFAR10 and STL10. $m = 64$ for CelebA and ImageNet

## A.3 PRETRAINING OF IMAGENET CLASSIFIERS

Both VGG19 and Resnet18 networks are trained with SGD with fixed $10^{-1}$ learning rate, 0.9 momentum term, and weight decay set to $5 \times 10^{-4}$. We pick models with best *top 1* accuracy on the validation set over 100 epochs of training; 29.14% for VGG19 (image size 32×32), and 39.63% for Resnet18 (image size 32×32). When training the classifiers we use random cropping and random horizontal flipping for data augmentation. When using VGG19 and Resnet18 as feature extractors in GFMN, we use features from the output of each ReLU that follows a conv. layer, for a total of 16 layers for VGG and 17 for Resnet18.

## A.4 QUANTITATIVE EVALUATION METRICS

We evaluate our models using two quantitative metrics: Inception Score (IS) (Salimans et al., 2016) and Fréchet Inception Distance (FID) (Heusel et al., 2017). We followed the same procedure used in previous work to calculate IS (Salimans et al., 2016; Miyato et al., 2018; Ravuri et al., 2018). For each trained generator, we calculate the IS for randomly generated 5000 images and repeat this procedure 10 times (for a total of 50K generated images) and report the average and the standard deviation of the IS.

We compute FID using two sample sizes of generated images: 5K and 50K. In order to be consistent with previous works (Miyato et al., 2018; Ravuri et al., 2018) and be able to directly compare our quantitative results with theirs, the FID is computed as follows:

- CIFAR10: the statistics for the real data are computed using the 50K training images. This (real data) statistics are used in the FID computation of both 5K and 50K samples of generated images. This is consistent with both Miyato et al. (2018) and Ravuri et al. (2018) procedure to compute FID for CIFAR10 experiments.

- STL10: when using 5K generated images, the statistics for the real data are computed using the set of 5K (labeled) training images. This is consistent with the FID computation of Miyato et al. (2018). When using 50K generated images, the statistics for the real data are computed using a set of 50K images randomly sampled from the unlabeled STL10 dataset.

FID computation is repeated 3 times and the average is reported. There is very small variance in the FID results.

## A.5 GENERATOR INITIALIZATION

The use a of pretrained decoder $D$ to initialize the generator gives a good boost in performance when the feature extractor is the decoder from a pretrained autoencoder (see Sec. 4.2.1). In Fig. 7, we show a visual comparison that demonstrates the effect of using the pretrained decoder $D$ to initialize the generator. The generators for the images in the first row (7a, 7b, 7c) were not initialized, while the generators for the images in the second row (7d, 7e, 7f) were initialized. For the three datasets, we can see a significant improvement in image quality when the generator is initialized with the decoder. We use the Laplacian pyramid loss to train CIFAR10 and CelebA AEs. However, L2 loss gives results almost as good as Lap1 loss.

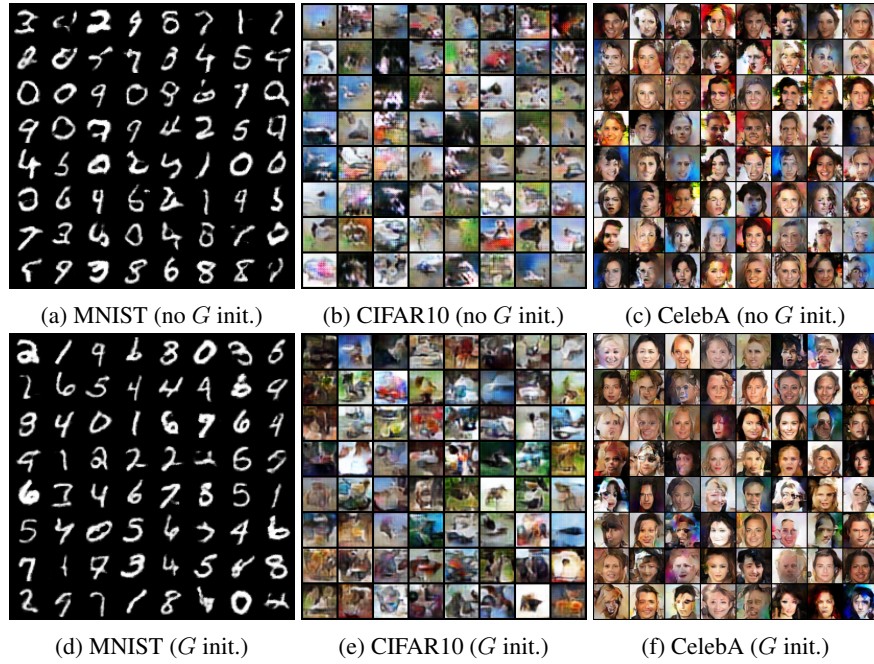

(a) MNIST (no $G$ init.)  (b) CIFAR10 (no $G$ init.)  (c) CelebA (no $G$ init.)

(d) MNIST ($G$ init.)  (e) CIFAR10 ($G$ init.)  (f) CelebA ($G$ init.)

Figure 7: Generated samples from GFMN using pretrained encoder as feature extractor. Visual comparison of models generated without (top row) and with (bottom row) initialization of the generation.

## A.6 CROSS-DOMAIN AUTOENCODER FEATURE EXTRACTORS

The experimental results in Sec. 4.2.2 demonstrate that cross-domain feature extractors based on DCNNs classifiers are very successful. For instance, we successfully used a VGG19 pretrained on ImageNet to train a GFMN generator for the CelebA dataset. Here, we investigate the impact of the pretraining of autoencoder-based feature extractors in a cross-domain setting. The objective is to further verify if GFMN is dependent on pretraining the autoencoder feature extractors on the same training data where the generator is trained. In Tab. 6, we show the results for different combinations of cross-domain feature extractors and $G$ initialization for STL10 and CIFAR10. The subscript indicates which dataset was used for pretraining. We can see in Tab. 6 that, CIFAR10 using $E_{\text{STL}}$ and $D_{\text{STL}}$ has similar performance (even better IS) to using $E_{\text{CIFAR}}$ and $D_{\text{CIFAR}}$. There is a performance drop when using $E_{\text{CIFAR}}$ and $D_{\text{CIFAR}}$ to train a STL10 generator. We believe this drop is related to the training set size. STL10 contains 100K (unlabeled) training examples while CIFAR10 contains 50K training images.

## A.7 IMPACT OF THE NUMBER OF LAYERS USED FOR FEATURE EXTRACTION

Figure 8 shows generated images from generators that were trained with a different number of layers employed to feature matching. In all the results in Fig.8, the VGG19 network was used to perform feature extraction. We can see a significant improvement in image quality when more layers are used.

Table 6: Cross-domain results for STL10 and CIFAR10 autoencoder features.

| $G$ Train. Data | Feat. Extractor | $G$ Initialization | IS | FID (5K) |
|---|---|---|---|---|
| STL10 | $E_{\text{STL}}$ | $D_{\text{STL}}$ | 4.74±0.12 | 69.80 |
| | $E_{\text{CIFAR}}$ | $D_{\text{CIFAR}}$ | 4.05±0.08 | 74.80 |
| CIFAR10 | $E_{\text{CIFAR}}$ | $D_{\text{CIFAR}}$ | 4.47±0.08 | 71.47 |
| | $E_{\text{STL}}$ | $D_{\text{STL}}$ | 4.53±0.09 | 73.74 |

Better results are achieved when 11 or more layers are used, which corroborates the quantitative results in Tab. 2.

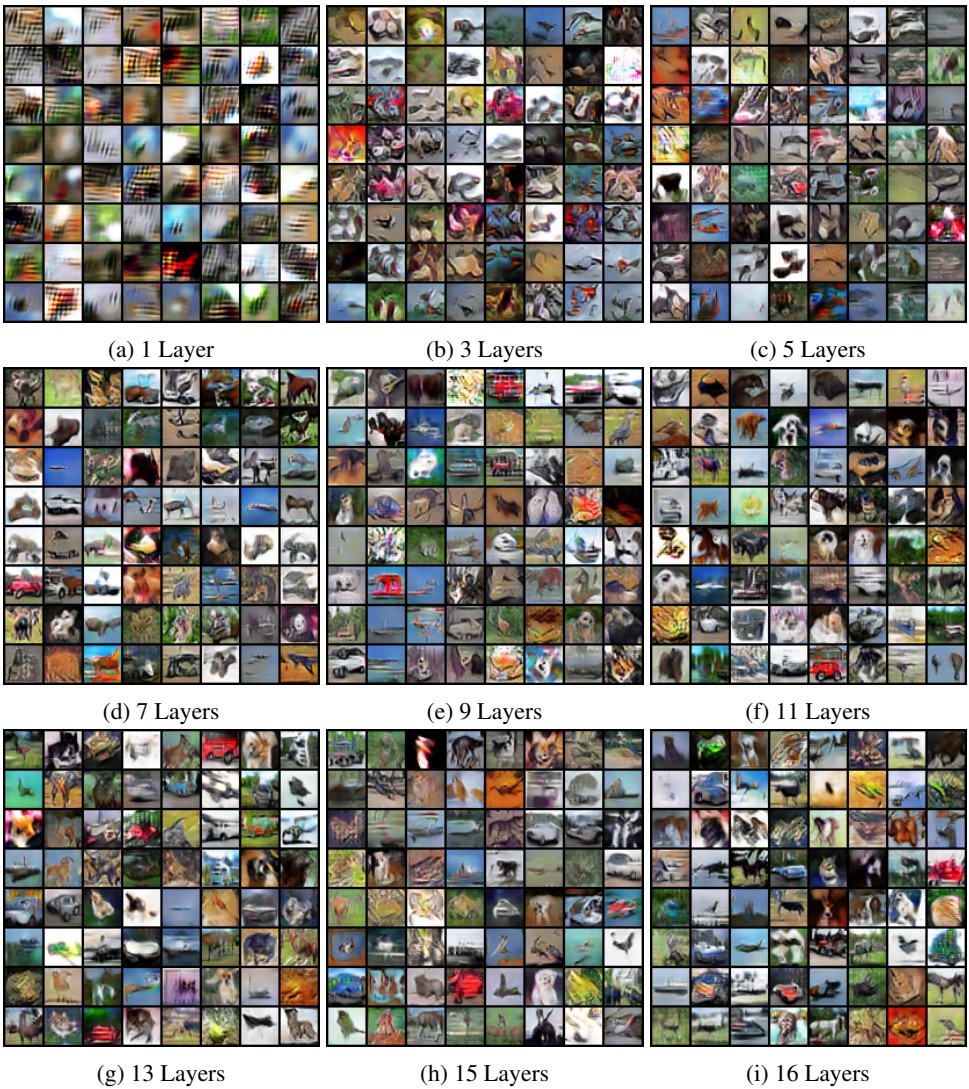

(a) 1 Layer (b) 3 Layers (c) 5 Layers

(d) 7 Layers (e) 9 Layers (f) 11 Layers

(g) 13 Layers (h) 15 Layers (i) 16 Layers

Figure 8: Generated images from GFMN trained with a different number of VGG19 layers for feature extraction.

## A.8 PRETRAINED GENERATOR/DISCRIMINATOR IN WGAN-GP

The objective of the experiments presented in this section is to evaluate if WGAN-GP can benefit from DCNN classifiers pretrained on ImageNet. In the experiments, we used a WGAN-GP architecture

where: (1) the discriminator is a VGG19 or a Resnet18; (2) the discriminator is pretrained on ImageNet; (3) the generator is pretrained on CIFAR10 through autoencoding. Although we tried different hyperparameter combinations, we were not able to successfully train WGAN-GP with VGG19 or Resnet18 discriminators. Indeed, the discriminator, being pretrained on ImageNet, can quickly learn to distinguish between real and fake images. This limits the reliability of the gradient information from the discriminator, which in turn renders the training of a proper generator extremely challenging or even impossible. This is a well-known issue with GAN training (Goodfellow et al., 2014) where the training of the generator and discriminator must strike a balance. This phenomenon is covered in (Arjovsky et al., 2017) Section 3 (illustrated in their Figure 2) as one motivation for work like Wassertein GANs. If a discriminator can distinguish perfectly between real and fake early on, the generator cannot learn properly and the min/max game becomes unbalanced, having no good discriminator gradients for the generator to learn from, producing degenerate models. Fig. 9 shows some examples of images generated by the unsuccessfully trained models.

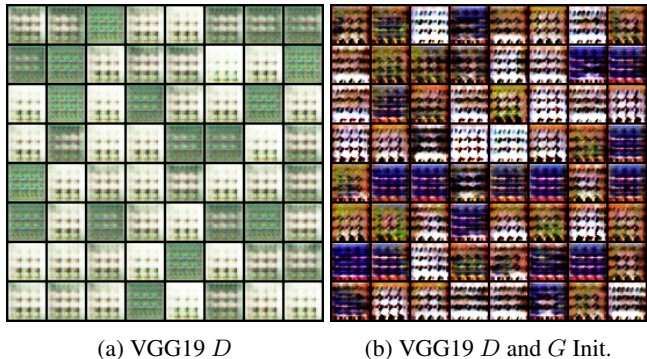

(a) VGG19 $D$  (b) VGG19 $D$ and $G$ Init.

Figure 9: Generated images by WGAN-GP with pretrained VGG19 as a discriminator.

A.9  IMPACT OF ADAM MOVING AVERAGE FOR VGG19 FEATURE EXTRACTOR.

In this appendix, we present a comparison between the simple moving average (MA) and ADAM moving average (AMA) for the case where VGG19 ImageNet classifier is used as a feature extractor. This experiment uses a minibatch size of 64. We can see in Fig. 10 that AMA has a very positive effect in the quality of generated images. GFMN trained with MA produces various images with some sort of crossing line artifacts.

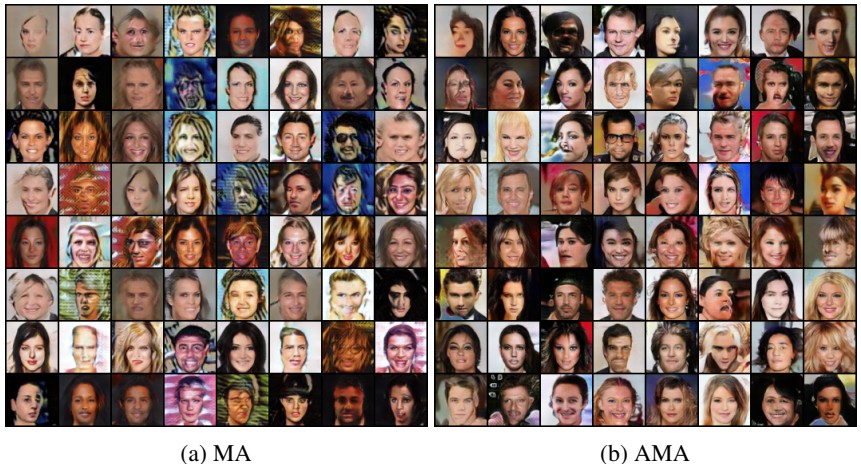

(a) MA  (b) AMA

Figure 10: Generated images from GFMN trained with either simple moving average (MA) or Adam moving average (AMA). VGG19 ImageNet classifier is used as feature extractor.

## A.10 Visual Comparison between GFMN and GMMN Generated Images.

Figure 11 shows a visual comparison between images generated by GFMN (Figs. 11a and 11b) and Generative Moment Matching Networks (GMMN) (Figs. 11c and 11d). GMMN (Li et al., 2015) generated images were obtained from Li et al. (2017).

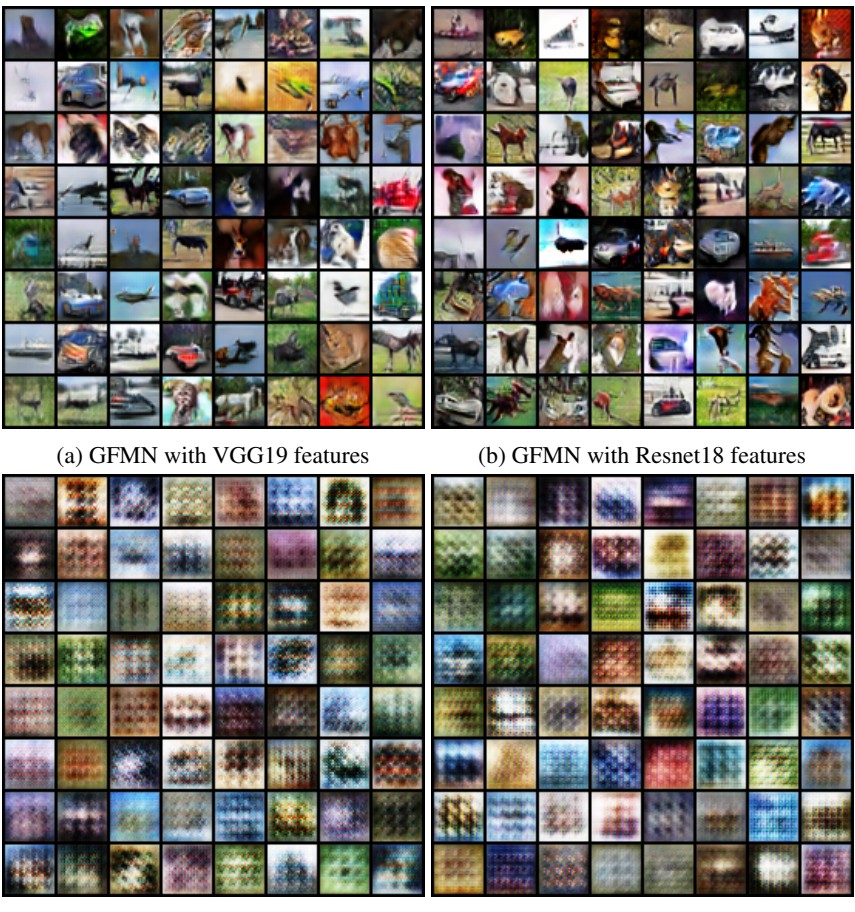

(a) GFMN with VGG19 features      (b) GFMN with Resnet18 features

(c) GMMN - Matching on data space      (d) GMMN+AE - Matching on AE space

Figure 11: Generated images from GFMN (11a and 11b) and GMMN (11c and 11d). GMMN images were obtained from Li et al. (2017).

## A.11 Sampling from the Pretrained Decoder vs. Sampling from GFMN models.

In this appendix, we present a visual comparison between images generated by sampling directly from decoders of pretrained autoencoders, and images generated by GFMN generators which were initialized by the decoders. In Fig 12, the images in the top row (Figs. 12a, 12b and 12c) were generated by decoders trained using CelebA, CIFAR10 and STL10, respectively. Images in the bottom row (Figs. 12d, 12e and 12f) were generated using GFMN generators that were initialized with CelebA, CIFAR10 and STL10 decoders, respectively. As expected, sampling from the decoder produces completely noisy images because the latent space is not aligned with the prior distribution $p_z$. GFMN uses the pretrained decoder as a better starting point and learns an effective implicit generative model, as we can see in Figs. 12d, 12e and 12f. Nevertheless, as demonstrated in Sec. 4.2.1, GFMN is also very effective without generator initialization, specially when using VGG19/Resnet18 feature extractors. Therefore, generator initialization is an interesting positive feature of GFMN, but not an essential aspect of the method.

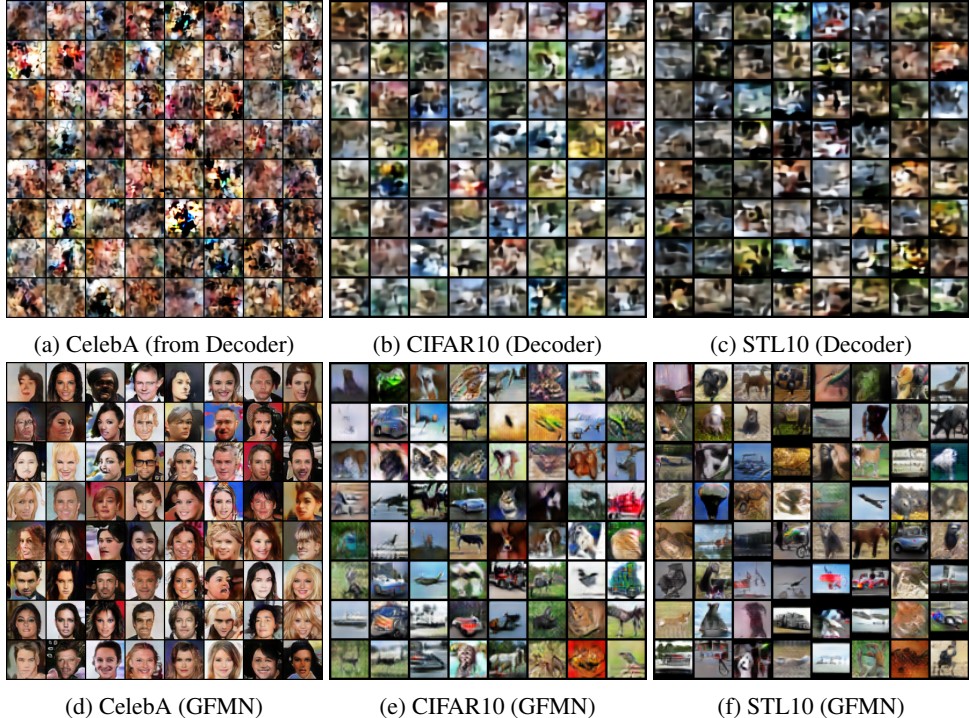

(a) CelebA (from Decoder)   (b) CIFAR10 (Decoder)   (c) STL10 (Decoder)

(d) CelebA (GFMN)   (e) CIFAR10 (GFMN)   (f) STL10 (GFMN)

Figure 12: Random samples from the decoders used to initialize GFMN generators (Figs. 12a, 12b and 12c), and from the generators after GFMN training (Figs. 12d, 12e and 12f) .

## A.12   IMPACT OF USING GLOBAL MEAN FEATURES VS. MINIBATCH-WISE MEAN FEATURES OF THE REAL DATA.

In this appendix, we assess whether GFMN is impacted by computing the mean features of the real data in a minibatch-wise fashion (computed in the minibatch and carried along with a moving average) vs. computing in a global manner (pre-computing the mean features using the whole training dataset, and keeping it fixed during the training). Note that this is for the real data only, for the fake data, we need to use a moving average because its mean features change a lot throughout the training process. In Fig. 13, we show generated images from GFMN trained with either simple Moving Average (MA) (13a, 13b, 13d and 13e) or Adam Moving Average (AMA) (13c and 13f). For MA, two minibatch sizes (mbs) are used: 64 and 128. Images in the top row were generated by models that perform feature matching using the minibatch-wise mean of the features from the real data, while the models that generated the images in the bottom row used global mean (gm) features computed in the whole CelebA dataset. We can see in Fig. 13 that using the global mean features does not improve the performance when training with MA, and also does not seem to have any impact when training with AMA.

## A.13   AUTOENCODER FEATURES VS. VGG19 FEATURES FOR CELEBA.

In this appendix, we present a comparison in image quality for autoencoder features vs. VGG19 features for the CelebA dataset. We show results for both simple moving average (MA) and ADAM moving average (AMA), for both cases we use a minibatch size of 64. In Fig. 14, we show generated images from GFMN trained with either VGG19 features (top row) or autoencoder (AE) features (bottom row). We show images generated by GFMN models trained with simple moving average (MA) and Adam moving average (AMA). We can note in the images that, although VGG19 features are from a cross-domain classifier, they lead to much better generation quality than AE features, specially for the MA case.

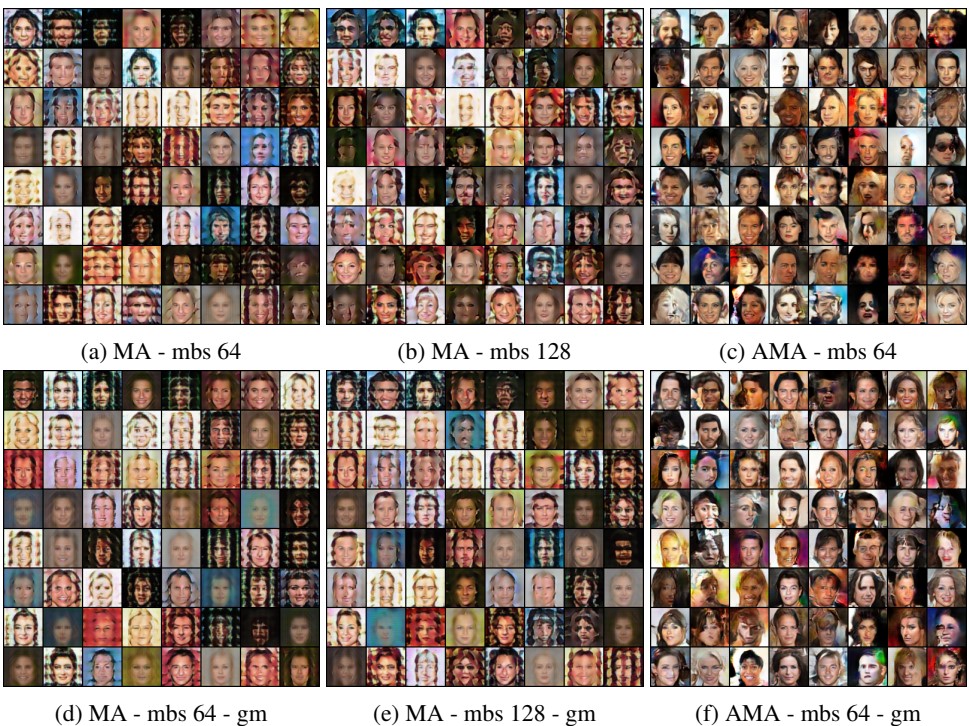



(a) MA - mbs 64     (b) MA - mbs 128     (c) AMA - mbs 64

(d) MA - mbs 64 - gm     (e) MA - mbs 128 - gm     (f) AMA - mbs 64 - gm



Figure 13: Generated images from GFMN trained with either simple Moving Average (MA) (13a, 13b, 13d and 13e) or Adam Moving Average (AMA) (13c and 13f). For MA, two minibatch sizes (mbs) are used: 64 and 128. Images in the top row were generated by models that perform feature matching using the minibatch-wise mean of the features from the real data, while the models that generated the images in the bottom row used global mean (gm) features computed in the whole CelebA dataset.

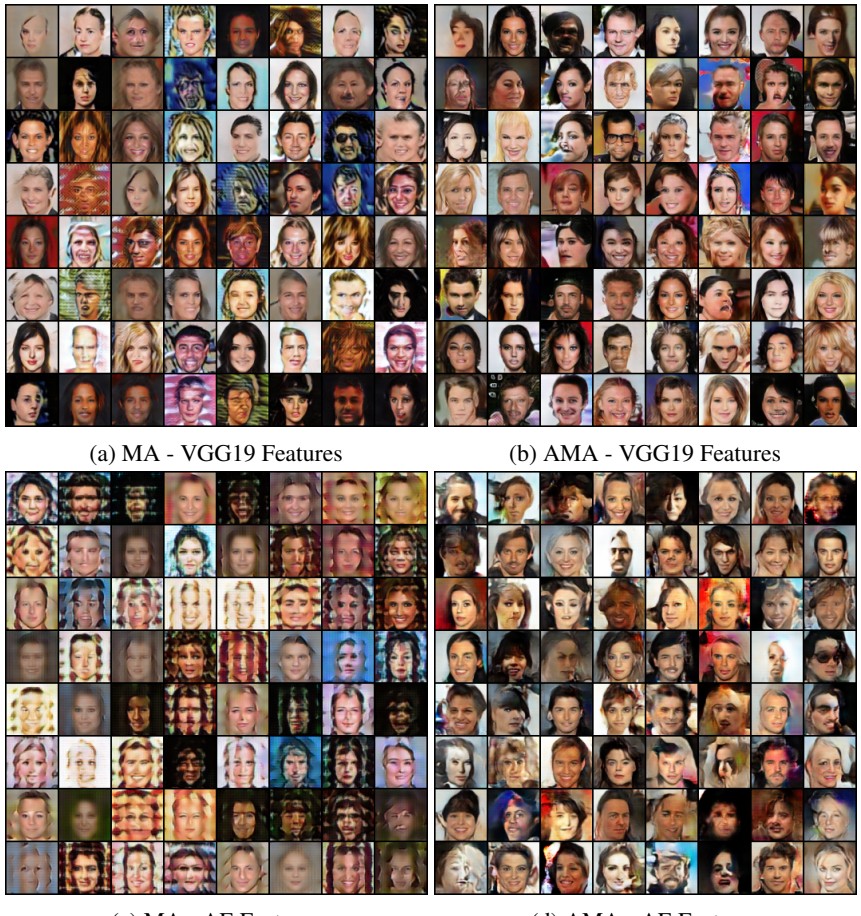

(a) MA - VGG19 Features        (b) AMA - VGG19 Features

(c) MA - AE Features        (d) AMA - AE Features

Figure 14: Generated images from GFMN trained with either VGG19 features (top row) or autoencoder (AE) features (bottom row). We show images generated by GFMN models trained with simple moving average (MA) and Adam moving average (AMA). Although VGG19 features are from a cross-domain classifier, they perform much better than AE features, specially for the MA case.

