# OpenReview forum: "Generative Feature Matching Networks"
_ICLR.cc/2019/Conference_

### Official Review · AnonReviewer2 · 2018-10-29
**Interesting results supported by experiments**

**Rating:** 6
**Confidence:** 3

**Review:**

The paper introduces Generative Feature Matching Networks (GFMNs) which is a non-adversarial approach to train generative models based on feature matching. GFMN uses pretrained neural networks such as Autoencoders (AE) and Deep Convolutional Neural Networks (DCNN) to extract features. Equation (1) is the proposed loss function for the generator network. In order to avoid big mini-batches, the GFMN performs feature matching with ADAM moving average.The paper validates its proposed approach with several experiments applied on benchmark datasets such as CIFAR10 and ImageNet.

The paper is well-written and straight-forward to follow. The problem is well-motivated by fully discussing the literature and the proposed method is clearly introduced. The method is then validated using several different experiments.

Typos:
** Page 1 -- Paragraph 3 -- Line 8: "(2) mode collapsing in not an issue"

---

> ### Author Response · Authors · 2018-11-13
> **Answer to AnonReviewer2**
>
>
> We would like to thank the reviewer for the positive feedback.  We have done a careful proof reading on the paper and addressed the typos pointed by the reviewer. Additionally, we have greatly improved and expanded sections 2.4 and 4.2.3 and added the new section 4.3 as well as two new appendices (A.9 and A.10). In the post destined to all the reviewers, we give more details about the main changes in the new version of the paper.
>
> We would like to reinforce that our paper presents a solid work backed by an extensive number of experiments and discussions. Moreover, as we present a method that evidence the power of pretrained DCNN representations for training generative models, we believe that our work is a perfect fit for ICLR and of big interest for its community.
>
> Please let us know if you need any additional clarification which would help you to better evaluate the work and increase the overall rating.

---

> ### Author Response · Authors · 2018-12-12
> **No feedback after paper revision.**
>
> Dear reviewer,
>
> We have incorporated your typo corrections in our revision of the paper, along with some additional discussions, ablation experiments and more details about Adam Moving average based on other reviewers' feedback. Could you please let us know if this helped you in better assessing our paper and in influencing your rating of the paper?
>
> Best,
> Authors

---

> > ### Comment · AnonReviewer2 · 2018-12-12
> > **No change in assessment**
> >
> > Thanks for updating the paper. Considering this version, my review still stays the same as a Weak Accept.

---

> > > ### Author Response · Authors · 2018-12-12
> > > **Reply to "No change in assessment"**
> > >
> > >
> > > Dear Reviewer,
> > >
> > > Could you please elaborate about the basis for rating our paper with 6? Your review only mentions positive things about the paper and points out one typo (which was corrected), and do not provide any reasons and justifications for giving a low score of 6 (weak accept). What do you think would have been acceptable for you to reach a rating 7 or higher?
> > >
> > > Our paper presents a solid work backed by strong experimental results, lots of ablation studies and discussions. From your review, it is hard to know what would be missing in our paper that would change your point of view.
> > >
> > > Best,
> > > the authors.

---

### Official Review · AnonReviewer1 · 2018-11-05

**Rating:** 6
**Confidence:** 3

**Review:**

This paper consists of two contributions: (1) using a fixed pre-trained network as a discriminator in feature matching loss ((Salimans et al., 2016). Since it's fixed there is no GAN-like training procedure. (2) Using "ADAM"-moving average to improve the convergency for the feature matching loss.

The paper is well written and easy to follow but it lack of some intuition for the proposed approach. There also some typo, e.g. "quiet" -> quite. Overall, it's a combination of several published method so I would expect a strong performance/analysis on the experimental session.

Detailed Comment:

For contribution (1):

The proposed method is very similar to (Li et al. (2015)) as the author pointed out in related work besides this work map to data space directly. Is there any intuition why this is better?

The proposed loss (the same as (Salimans et al., 2016)) only try to matching first-order momentum. So I assume it is insensitive to higher-order statistics. Does it less successful at producing samples with high visual fidelity?

For contribution (2):

"one would need big mini-batches which would result in slowing down the training." why larger batch slowing down the training? Is there any qualitative results? Based recent paper e.g. big gan, it seem the model can benefit a lot from larger batch. In the meanwhile, even larger batch make it slower to converge, it can improve throughput.

Again, can the author provide some intuition for these modification? It's also unclear to me what is ADAM(). Better to link some equation to the original paper or simply write down the formulation and give some explanation on it.

For experiments:

I'm not an expert to interpret experimental results for image generation. But overall, the results seems not very impressive. Given the best results is using ImageNet as a classifier, I think it should compare with some semi-supervised image generation paper.

For example, for CIFAR results, it seems worse than (Warde-Farley & Bengio, 2017), Table 1, semi-supervised case. If we compare unsupervised case (autoencoder), it also seems a lot worse.

Appendix A.8 is very interesting / important to apply pre-trained network in GAN framework. However, it only say failed to train without any explanation.

I think even it just comparable with GAN, it is interesting if there is no mode collapsing and easy to train. However, it has no proper imagenet results (it has a subset, but only some generated image shows here).

In summary, this paper provide some interesting perspectives. However, the main algorithms are very similar to some existing methods, more discussion could be used to compare with the existing literature and clarify the novelty of the current paper. The empirical results could also be made more stronger by including more relevant baseline methods and more systematic study of the effectiveness of the proposed approach. I tend to give a weak reject or reject for this paper.

---

Update post in AC discussion session.

---

> ### Author Response · Authors · 2018-11-13
> **Answer to AnonReviewer1 (Part 3/3)**
>
>
> Regarding experiments:
>
> Rev: “I think even it just comparable with GAN, it is interesting if there is no mode collapsing and easy to train. However, it has no proper imagenet results (it has a subset, but only some generated image shows here).” “The empirical results could also be made more stronger by including more relevant baseline methods and more systematic study of the effectiveness of the proposed approach.”
>
> We compare our results with Miyato (ICLR 2018), Bikowski (ICLR 2018) and Ravuri et al. (ICML 2018) which are very strong up-to-date baselines (all published in 2018) and were the state-of-the-art results by the time we submitted the paper.
>
> Regarding ImageNet results, note that in this paper we do not propose to perform conditional generation. All (very recent) papers reporting IS/FID for ImageNet perform conditional generation. Our ImageNet results should be compared with (Ravuri et al., 2018) for the Daisy portion, and (Salimans et. al, 2016; Zhao et al., 2017) for the ImageNet dogs portion. Again, it is not fair to compare our results with the ones from large-scale experiments or with systems that explicitly model conditional generation.
>
> Note that the focus of this paper is to demonstrate that we can train effective generative models without adversarial training by employing frozen pretrained neural networks. We selected benchmarks that have been used for the last three/four years by the community on gen. models: CIFAR10, CelebA, MNIST, STL10. We performed an extensive number of experiments, reported quantitative results using two metrics (IS and FID)  and systematically assessed multiple aspects of the proposed approach:
> (1)  We checked the advantage of AMA vs MA;
> (2)  demonstrated the correlation of loss vs. image quality;
> (3)  evaluated different methods for pretraining the feature extractor (autoencoding, classification);
> (4)  checked different architectures for the feat. extractor (DCGAN, VGG19, Resnet18);
> (5)  assessed the impact of the number of features/layers used;
> (6)  evaluated in-domain and cross domain feature extractors;
> (7)  tested the benefit of initializing the generator;
> (8)  evaluated the joint use of multiple feature extractors (VGG19 + Resnet18) for training the same generator;
> (9)   performed experiments with WGAN-GP initialized with VGG19/Resnet18;
> (10) presented results for different portions of ImageNet;
> (11) presented visual comparison of images generated between GFMN and GMMN;
> (12) compared our results with state-of-the-art methods.
> Moreover, the improved Sec. 4.2.3 now contains even more experiments and discussions regarding the advantages of using AMA.
>
> Finally, we would like to reinforce that our paper presents a solid work backed by an extensive number of experiments and discussions. Moreover, as we present a method that provides evidence for the power of pretrained DCNN representations for learning generative models, we believe our work is a perfect fit for ICLR and is of great interest for its community.
>
> Please let us know if you need any additional clarification which would help you to better evaluate the work and increase the overall rating.

---

> ### Author Response · Authors · 2018-11-13
> **Answer to AnonReviewer1 (Part 2/3)**
>
>
> Regarding contribution (2):
>
> Rev: ""one would need big mini-batches which would result in slowing down the training." why larger batch slowing down the training? Is there any qualitative results? Based recent paper e.g. big gan, it seem the model can benefit a lot from larger batch. In the meanwhile, even larger batch make it slower to converge, it can improve throughput. "
> Rev: “Again, can the author provide some intuition for these modification? It's also unclear to me what is ADAM(). Better to link some equation to the original paper or simply write down the formulation and give some explanation on it.”
>
> We have significantly extended and improved Sec. 2.4 which describes our proposed Adam Moving Average (AMA). We have included more details about the motivation and intuition behind the proposed method. We have also added ADAM equations so that the description is self-contained.
>
>
> In summary, not training time, but memory usage is the main motivation for using moving averages. When using images larger than 32x32 and DCNNs that produce millions of features, this can easily result in memory issues. Moving average is a strategy to alleviate this problem.
> We propose the ADAM Moving Average (AMA) to further promote stable training when using small minibatches. The main advantage of AMA over simple moving average (MA) is its adaptive first order and second order moments that ensure a stable estimation of the moving averages. In fact, this is a non-stationary estimation since the mean of the generated data changes in the training, and it is well known that ADAM optimizer works well for such online and non-stationary losses (Kingma & Ba, 2015).
>
> We have improved and expanded “Sec 4.2.3.”, which now contains more detailed experiments to further demonstrate the advantage of AMA over MA. Sec 4.2.3 now brings more discussions about AMA vs MA experimental results. Additionally, we have added a new Appendix “A.9”, which presents experimental results that also indicate the advantage of AMA over AM when VGG19 feature extractor is employed.
>
>
> Regarding experiments:
>
> Rev: “Given the best results is using ImageNet as a classifier, I think it should compare with some semi-supervised image generation paper. For example, for CIFAR results, it seems worse than (Warde-Farley & Bengio, 2017), Table 1, semi-supervised case. If we compare unsupervised case (autoencoder), it also seems a lot worse.”
>
> Comparing our results with the ones from semi-supervised approaches is not fair because we do not use labels from the target dataset, CIFAR10. Nevertheless, the difference between the result of our best system for CIFAR10 and the semi-supervised system reported in Table 1 of (Warde-Farley & Bengio, 2017) is not statistically significant: Inception Score (IS) of 7.99 (ours) vs 8.06 (theirs). This is actually an impressive result in itself, because our system does not use labels from CIFAR10 and our critic (feature extractor) is not updated during the training of the generator.
>
> Although you might argue that the best results from GFMN are obtained with feature extractors that were trained in a supervised manner (classifiers), note that:
> (1) We have also tried to initialize the discriminator of GANs with a pretrained ImageNet classifier, but it failed to train (more on this in next answer).
> (2) The accuracy of the classifier does not seem to be a very important factor for generating good features (VGG19 classifier produces better features although it is less accurate than Resnet18, see Appendix A.3); we are very confident that GFMN will also achieve state-of-the-art results when trained with features from classifiers pretrained using unsupervised methods such as the one recently proposed by Caron et al. (2018). This is something that can be explored in future works.
>
> Rev: “Appendix A.8 is very interesting / important to apply pre-trained network in GAN framework. However, it only say failed to train without any explanation.”
>
> We have expanded Appendix A.8 with additional details/explanations on the reasons why WGAN-GP fails when we use a pretrained VGG19/Resnet18 to initialize the discriminator. In short, the discriminator, being pretrained on ImageNet, can quickly learn to distinguish between real and fake images. This limits the reliability of the gradient information from the discriminator, which in turn renders the training of a proper generator extremely challenging or even impossible. This is a well-known issue with GAN training (Goodfellow et al., 2014) where the training of the generator and discriminator must strike a balance. This phenomenon is covered in (Arjovsky et al., 2017) Section 3 (illustrated in their Figure 2) as one motivation for works on Wasserstein GANs.

---

> ### Author Response · Authors · 2018-11-13
> **Answer to AnonReviewer1 (Part 1/3)**
>
> We would like to thank the reviewer for the detailed questions, comments and suggestions. We believe they have helped us improve the quality of the paper. We have done substantial changes in the text in order to address your questions/comments/suggestions.
> In the post destined to all reviewers, we give a detailed description of the main changes in the new version of the paper. We kindly ask the reviewer to take a look at the post as well as the new version of the paper.
> Please see below our answers for your questions/comments.
>
> Regarding contribution (1):
>
> Rev: “the main algorithms are very similar to some existing methods”
>
> Please note that all MMD methods are similar in the sense that they all perform moment matching. Using the same line of argumentation, you would also argue that all papers presenting new GANs are very similar to each other because all of them use the same adversarial strategy. One of the key points that makes our work unique is the demonstration that we can use pretrained neural networks as a powerful kernel function that allows robust and effective moment matching, a result that has never been demonstrated before. Throughout an extensive number of experiments with multiple datasets and different feature extractor architectures we demonstrate the robustness and effectiveness of the method. Moreover, to the best of our knowledge, this is the first work to present state-of-the-art results for CIFAR-10 and STL10 using a non-adversarial method that moves away completely from the problematic min/max game of GANs, which is an impressive result.
>
>
> Rev: “The proposed method is very similar to (Li et al. (2015)) as the author pointed out in related work besides this work map to data space directly. Is there any intuition why this is better?” “… more discussion could be used to compare with the existing literature and clarify the novelty of the current paper.”
>
> We have included a new section “4.3 Discussion” which better details the differences and advantages of GFMN over GANs and other MMD approaches. In short:
>
> (1) Compared to GANs, we achieve better results than the state-of-the-art SN-GAN while completely avoiding the problematic min/max game. Our method has stable training and no mode-collapsing.
>
> Compared to other MMD approaches:
> (2) we present far better quantitative results than GMMN (Li et al., 2015). Additionally, the new Appendix A.10 shows a visual comparison between GMMN and GFMN results. The main reason why GFMN results are significantly better than GMMN is because GFMN uses a strong, robust kernel function (a pretrained DCNN), which, together with our AMA trick, allows a stable and effective training with small minibatches. On the other hand, the Gaussian kernel used in GMMN requires a very large minibatch size in order to work well, which is impractical due to memory limitations and computational cost;
>
> (3) Compared to recent adversarial MMD methods (Li et al., 2017; Bikowski et al., 2018) GFMN presents significantly better results while avoiding the problematic min/max game;
>
> (4) GFMN achieves similar results to the Method of Learned Moments (MoLM) (Ravuri et al., 2018), while using 50x less moments/features to perform matching. In other words, while MoLM can be used in large-scale environments only, GFMN can be used in single GPU environments achieving the same or better performance.
>
>
> Rev: “The proposed loss (the same as (Salimans et al., 2016)) only try to matching first-order momentum. So I assume it is insensitive to higher-order statistics. Does it less successful at producing samples with high visual fidelity?”
>
> It is not quite correct to say that our proposed approach is insensitive to higher-order statistics. Note that using a pretrained DCNN to extract features is equivalent to use a highly non-linear kernel function to map the data into a very high dimensional space (hundreds of thousands of dimensions in our case). What we show in our experiments is that, in this very high dimensional space, matching first order statistics is already enough to achieve state-of-the-art results. We demonstrate empirically that our strategy is more efficient and effective than methods such as GMMN, which use a Gaussian kernel to match all the moments. The problem with using a Gaussian kernel is that it requires a very large minibatch size in order to produce good estimates, which is not feasible in practice (Li et al., 2017).
>
> Regarding visual fidelity, our method produces images that are better or on par with other systems that use similar generator architectures, similar computational resources and do not use conditional generation. For instance, if you compare the quality of the (unconditional) generated images using ImageNet Dogs, you will see that our results are much better than (Salimans et. al, 2016) and (Zhao et al., 2017). It is not fair to compare our results with large-scale experiments or with systems that explicitly model conditional generation.

---

> ### Author Response · Authors · 2018-12-12
> **No feedback after paper revision.**
>
>
> Dear Reviewer,
>
> We have incorporated your feedback in our revision of the paper and added detailed clarifications for all your questions/comments in our reply. Could you please let us know if this helped you in better assessing our paper and in influencing your rating of the paper?
>
> Best,
> Authors

---

### Official Review · AnonReviewer3 · 2018-11-06
**Review for "Generative Feature Matching Networks"**

**Rating:** 6
**Confidence:** 3

**Review:**

The paper proposes a non-adversarial feature matching generative model (GFMN). In feature matching GANs, the discriminator extract features that are employed by the generator to match the real data distribution. Through the experiments, the paper shows that the loss function is correlated with the generated image quality, and the same pretrained feature extractor (pre-trained on imagenet) can be employed across a variety of datasets. The paper also discusses the choice of pretrained network or autoencoder as the feature extractor. The paper also introduces an ADAM-based moving average. The paper compares the results with on CIFAR10 and STL10 with a variety of recent State-of-the-art approaches in terms on IS and FID.

+ The paper is well written and easy to follow. - However, there are some typos that should be addressed. Such as:
“The decoder part of an AE consists exactly in an image generator ”
“Our proposed approach consists in training G by minimizing”
“Different past work have shown” -> has
“in Equation equation 1 by”
“have also used” better to use the present tense.

+ It suggests a non-adversarial approach to generate images using pre-trained networks. So the training is easier and the quality of the generated images, as well as the FID and IS, are still comparable to the state-of-the-art approaches.

---

> ### Author Response · Authors · 2018-11-13
> **Answer to AnonReviewer3**
>
>
> We would like to thank the reviewer for the positive feedback.  We have done a careful proof reading on the paper and addressed the typos pointed by the reviewer. Additionally, we have greatly improved and expanded sections 2.4 and 4.2.3 and added the new section 4.3 as well as two new appendices (A.9 and A.10). In the post destined to all the reviewers, we give more details about the main changes in the new version of the paper.
>
> We would like to reinforce that our paper presents a solid work backed by an extensive number of experiments and discussions. Moreover, as we present a method that evidence the power of pretrained DCNN representations for training generative models, we believe that our work is a perfect fit for ICLR and of big interest for its community.
>
> Please let us know if you need any additional clarification which would help you to better evaluate the work and increase the overall rating.

---

### Official Review · AnonReviewer4 · 2018-11-21
**-**

**Rating:** 6
**Confidence:** 4

**Review:**

This paper proposes to learn implicit generative models by a feature matching objective which forces the generator to produce samples that match the means of the data distribution in some fixed feature space, focusing on image generation and feature spaces given by pre-trained image classifiers.

On the positive side, the paper is well-written and easy to follow, the experiments are clearly described, and the evaluation shows the method can achieve good results on a few datasets. The method is nice in that, unlike GANs and the stability issues that come with them, it minimizes a single loss and requires only a single module, the generator.

On the other hand, the general applicability of the method is unclear, the novelty is somewhat limited, and the evaluation is missing a few important baselines. In detail:

1) The proposed objective was used as a GAN auxiliary objective in [Salimans et al., 2016] and further explored in [Warde-Farley & Bengio, 2017]. The novel bit here is that the proposed objective doesn’t include the standard GAN term (so no need for an adversarially-optimized discriminator), and the feature extractor is a fixed pre-trained classifier or encoder from an auto-encoder (rather than a discriminator).

2) The method only forces the generator’s sample distribution to match the first moment (the mean) of the data distribution. While the paper shows that this can result in a generator that produces reasonably good samples in practice, it seems like this may have happened due to a “lucky” artifact of the chosen pre-trained feature extractors. For example, a degenerate generator that produces a single image whose features exactly match the mean would be a global optimum under this objective, equally good as a generator that exactly matches the data distribution. Perhaps no such image exists for the chosen pre-trained classifiers, but it’s nonetheless concerning that the objective does nothing to prevent this type of behavior in the general case. (This is similar to the mode collapse problem that often occurs with GAN training in practice, but at least a GAN generator is required to exactly match the full data distribution to achieve the global optimum of that objective.)

3) It’s unclear why the proposed ADAM-based Moving Average (AMA) updates are appropriate for estimate the mean features of the data distribution. Namely, unlike EMA updates, it’s not clear that this is an unbiased estimator (I suspect it’s not); i.e. that the expectation of the resulting estimates is actually the true mean of the dataset features.  It’s therefore not clear whether the stated objective is actually what’s being optimized when these AMA updates are used.

4) Related to (3), an important baseline which is not discussed is the true fixed mean of the dataset distribution. In Sec. 2.4 (on AMA) it’s claimed that “one would need large mini-batches for generating a good estimate of the mean features...this can easily result in memory issues”, but this is not true: one could trivially compute the full exact dataset mean of these fixed features by accumulating a sum over the dataset (e.g., one image a time, with minibatch size 1) and then dividing the result by the number of images in the dataset. Without this baseline, I can’t rule out that the method only works due to its reliance on the stochasticity of the dataset mean estimates to avoid the behavior described in (2), or even the fact that the estimates are biased due to the use of ADAM as described in (3).

5) The best results in Table 3 rely on initializing G with the weights of a decoder pretrained for autoencoding. However, the performance of the decoder itself with no additional training from the GFMN objective is not reported, so it’s possible that most of the result relies on *autoencoder* training rather than feature matching to get a good generator. This explanation seems especially plausible due to the fact that the learning rate is set to a miniscule value (5*10^-6 for ADAM, 1-2 orders of magnitude smaller than typical values). Without the generator pretraining, the next best CIFAR result is an Inception Score of 7.67, lower than the unsupervised result from [Warde-Farley & Bengio, 2017] of 7.72.

6) It is misleading to call the results based on ImageNet-pretrained models “unconditional” -- there is plenty of overlap in the supervision provided by the labeled images of the much larger ImageNet to CIFAR and other datasets explored here. This is especially true given that the reported metrics (Inception Score and FID) are themselves based on ImageNet-pretrained classifiers. If the results were instead compared to prior work on conditional generation (e.g. ProGAN (Karras et al., 2017), which reports CIFAR IS of 8.56), there would be a clear gap between these results and the state of the art.

Overall, the current version of the paper needs additional experiments and clarifying discussion to address these issues.

=======================================

REVISION

Based on the authors' responses, I withdraw points 3-5 from my original review. Thanks to the authors for the additional experiments. On (3), I indeed misunderstood where the moving average was being applied; thanks for the correction. On (4), the additional experiment using the global mean features for real data convinces me that the method does not rely on the stochasticity of the estimates. (Though, given that the global mean works just as well, it seems like it would be more efficient and arguably cleaner to simply have that be the main method. But this isn't a major issue.) On (5), I misread the learning rate specified for "using the autoencoder features" as being the learning rate for autoencoder *pretraining*; thanks for the correction. The added results in Appendix 11 do show that the pretrained decoder on its own does not produce good samples.

My biggest remaining concerns are with points (2) and (6) from my original review.

On (2), I did realize that features from multiple layers are used, but this doesn't theoretically prevent the generator from achieving the global minimum of the objective by producing a single image whose features are the mean of the features in the dataset. That being said, the paper shows that this doesn't tend to happen in practice with existing classifiers, which is an interesting empirical contribution. (It would be nice to also see ablation studies on this point, showing the results of training against features from single layers across the network.)

On (6), I'm still unconvinced that making use of ImageNet classifiers isn't providing something like a conditional training signal, and that using such classifiers isn't a bit of an "unfair advantage" vs. other methods when the metrics themselves are based on an ImageNet classifier. I realize that ImageNet and CIFAR have different label sets, but most if not all of the CIFAR classes are nonetheless represented -- in a finer-grained way -- in ImageNet. If ImageNet and CIFAR were really completely unrelated, an ImageNet classifier could not be used as an evaluation metric for CIFAR generators. (And yes, I saw the CelebA results, but for this dataset there's no quantitative comparison with prior work, and qualitatively, if the results are as good as or better than the 3 year old DCGAN results, I can't tell.)

On the other hand, given that the approach relies on these classifiers, I don't have a good suggestion for how to control for this and make the comparison with prior work completely fair. Still, it would be nice to see acknowledgment and discussion of this caveat in a future revision of the paper.

Overall, given that most of my concerns have been addressed with additional experiments and clarification, and that the paper is well-written and has some interesting results from its relatively simple approach, I've raised my rating to above acceptance threshold.

---

> ### Author Response · Authors · 2018-11-24
> **Reply to AnonReviewer4 (Part 3 / 3):**
>
>
> 6) On ImageNet vs. CIFAR10; conditional generation; IS/FID metrics and experimental results:
>
> AUTHORS: The reviewer's concerns are based on misconceptions. Below we list some facts that will help to rule out the reviewer's concerns:
>
> (a) the label overlap between ImageNet and CIFAR10 is not direct at all. ImageNet uses fine grained labels, while CIFAR10 uses 10 labels only. For instance, while ImageNet has dozens of classes for dogs (one for each breed), CIFAR10 groups all dog images in one single class (see [3], Appendix F). Labeling the data in different ways produce quite different classification problems. Moreover, the images in CIFAR10 are not a subset of ImageNet since they have different resolutions. It is completely safe to say that CIFAR10 is an out-of-domain dataset with regard to ImageNet;
>
> (b) the reviewer is ignoring the fact that we also have good results for an extreme case of out-of-domain dataset: CelebA. There is no overlap between ImageNet and CelebA and the images from these two datasets are quite different. Nevertheless, the VGG19 classifier produces much better results than the autoencoder pretrained in CelebA. In the new Appendix 13, Fig 14, we show additional results that support this fact. These results are clear evidence that VGG19 classifiers is just a much better feature extractor than autoencoders, which is also the real reason for our boost in performance for CIFAR10 (and not "label overlapping"). There is a long literature (some cited in our paper) that show the effectiveness of VGG19 ImageNet classifier as feature extractor;
>
> (c) ImageNet classifiers, and in special VGG classifier, is the default choice for feature extraction in computer vision tasks. We are sure that  all the reviewers would have complained about our work if we have not used ImageNet-based feature extractors;
>
> (d) our use of VGG19/Resnet18 ImageNet classifiers have no impact in the used metrics for different reasons: (d.1) IS and FID are computed using the default tensorflow Inception model trained with images of size 299x299, while the classifier ( the feature extractor ) that we use in our experiments was trained using  images of size 32x32, as informed in Appendix A.3. Our ImageNet VGG19 classifier has top-1 accuracy of  29.14% while the Inception net has about top-1 accuracy of 79%. In summary, our classifier is completely different in many crucial aspects when compared to the Inception classifier used to compute IS and IFD; (d.2) we use the classifier as a feature extractor only, no log likelihoods from the classifier is used in our objective function;
>
> (e) we do not perform conditional generation. GAN-based methods that perform conditional generation use direct feedback from the labels in the form of log likelihoods from the discriminator (using the k+1 trick from Salimans et. al 2016) or from an auxiliary classifier. In the contrary, our generator is trained with a loss function that performs feature matching only, there is no feedback in form of a log likelihood from the labeled data. Our generator is agnostic to the labels (there is no one-hots concatenated to the noise) and  it is only distilling the knowledge of the multi-layer feature space, without explicitly taking advantage of any labeling that went to the training to this feature space;
>
> (f) ProGAN (Karras et al., 2017) uses a generator architecture that contains residual connections and is  deeper and more complex than the DCGAN-like architecture that we use in our CIFAR10 experiments. Their better performance is due mainly to the generator's architecture and the tricks used to train such an architecture. It is unfair to compare our results with the ones of a method that uses a bigger and more complex generator. We were careful and fair in trying to put in Table 3 the results for very recent and relevant work that use generator architecture that are similar to ours. By the way, the trick of progressively growing the generator can also be applied in GFMN and would likely increase our performance. But this experiment is completely out of the scope of the current paper;
>
> [3] Oliver et al. Realistic Evaluation of Deep Semi-Supervised Learning Algorithms. ArXiv 2018. https://arxiv.org/pdf/1804.09170.pdf

---

> > ### Comment · AnonReviewer4 · 2018-12-05
> > **-**
> >
> > Thank you for the detailed responses and additional experiments. I have updated my review above.

---

> > > ### Author Response · Authors · 2018-12-05
> > > **Further clarifications to AnonReviewer4**
> > >
> > >
> > > We would like to thank the reviewer for considering our response and updating your review. We really appreciate your feedback.
> > >
> > > Regarding your remaining  concerns:
> > >
> > > (2) Our paper already contains ablation experiments where we show the performance of generators that were trained with a different number of layers employed to feature matching.  Please check our experimental results in Table 2 (pg. 7) , Figs 2.d, 2.e, 2.f (pg. 6) and Appendix A.7 (Fig. 8, pg. 17). These ablation experiments are a clear evidence that training with just part of the features is sub-optimal, the best performance is  achieved when all the features are used. Training with the features of one single layer (independent of the layer) leads to very poor results. Although we present results for VGG19/Resnet18 classifiers only, the same behavior is true for the case of encoder feature extractor.
> > >
> > > (6) We agree on the difficulty of a comparison between GAN with learned discriminator and Imagenet based feature extractors. However, please note that we did additional experiments where we evaluated if WGAN-GP could benefit from initializing the discriminator with DCNN classifiers pretrained on ImageNet. This was the only strategy that we could come up with to somehow feed GANs with the same ImageNet feature extractors that GFMN uses. In Appendix A.8 (pgs. 17/18) we present experimental results and discussions regarding this experiment. Although we tried different hyperparameter combinations, we were not able to successfully train WGAN-GP with VGG19 or Resnet18 discriminators. It seems that the discriminator being pretrained on ImageNet, can quickly learn to distinguish between real and fake images. This limits the reliability of the gradient information from the discriminator, which in turn renders the training of a proper generator extremely challenging.
> > >
> > > Again, we believe that our use of ImageNet classifiers is fair game and our paper should not be penalized because of this. GFMN's good performance for CIFAR10 and CelebA is a good demonstration that it excels even with cross-domain feature extractors.

---

> > > > ### Comment · AnonReviewer4 · 2018-12-05
> > > > **-**
> > > >
> > > > > (2) Our paper already contains ablation experiments where we show the performance of generators that were trained with a different number of layers employed to feature matching.
> > > >
> > > > Indeed -- sorry, I'd forgotten about these results.

---

> > > > > ### Author Response · Authors · 2018-12-05
> > > > > **Thanks for the reply.**
> > > > >
> > > > > Thanks for your prompt answer and for acknowledging that. If there is any additional clarification that we can give in order to further improve your rating of the paper, please let us know.

---

> ### Author Response · Authors · 2018-11-24
> **Reply to AnonReviewer4 (Part 2 / 3):**
>
>
> 3) Is Adam moving average biased?
>
> AUTHORS:  There is a big misunderstanding here from the reviewer on what moving average we are talking about: It is not the average feature on the real dataset. It is the moving average of the difference of mean features between fake and real. As the generator is updated,  this moving average changes. Since this is a non stationary estimation (the generator changes during the training), Adam allows a better estimation of this moving average.
>
> Hence, it does not make sense to talk about biased or unbiased estimation here since the goal is not to estimate the mean of real data, but rather to get an estimate of the difference of means of real and fake, in a stochastic mini-batching context. In this non stationary context, Adam is known to excel.
>
> 4) Full Dataset mean experiment :
>
> AUTHORS:  Again this goes to the misunderstanding above: the moving average is on  "mean_fake - mean_real" and not only on the real dataset. As we update the  parameters of the generator,  we have a non stationarity difference of means to estimate . Even if mean_real was fixed to the full dataset, a simple moving average will not give us a good estimate.  That is why we introduced the Adam moving average to alleviate the non stationarity of the mean difference that we have to estimate. We added Appendix A.12, Figure 13,  that shows that if we have a mean for all the real dataset (global mean), one still needs moving average for the generator (fake) and that adam moving average outperforms the simple Moving average.
>
> The reasoning of the reviewer is flawed regarding no need of a large minibatch size: "... but this is not true: one could trivially compute the full exact dataset mean of these fixed features by accumulating a sum over the dataset (e.g., one image a time, with minibatch size 1) and then dividing the result by the number of images in the dataset".  Your reasoning is ignoring a fundamental problem that is, during training, one needs to also compute the mean feature of the generator (fake) data, compute the loss and back propagate the errors back to the generator. In order to perform the backpropagation, we need to keep in memory all the generated images and all the information about its forward step (for both G and the feature extractor), hence of course there is a memory problem.
>
> In summary, as we show in appendix A.12 Figure 13, yes one can have an estimate of the full mean of the real dataset and use the adam moving average on : "mean_fake - gm"  (gm =global average of real  ) and GFMN still succeeds, which confirms that we are effectively optimizing the feature matching cost function, and this rules out the reviewer's concerns.
>
>
> 5) On sampling from decoder vs. sampling from GFMN; G initialization and learning rate:
>
> AUTHORS:  We have added Appendix A.11 “Sampling from the Pretrained Decoder vs. Sampling from GFMN models” where we present a comparison between images generated by sampling from the decoder (with no additional training from GFMN) and images generated by GFMN. We can see in Fig 12 that, as expected, when sampling directly from decoders we obtain completely noisy images for the three datasets experimented: CelebA, CIFAR10 and STL10. Moreover, as you mentioned, our results without G initialization, 7.67 / 23.5 (IS/FID), are quite close to the results that use G initialization, 7.99 / 23.1 (IS/FID), which demonstrates that G initialization is responsible for a very small fraction of the result. Therefore, we can definitively rule out the reviwer's concern that “most of the result relies on *autoencoder* training rather than feature matching”. Generator initialization is an interesting positive feature of GFMN, but is far from being the essential component of the method.
>
> Regarding learning rate, please note that in the paper we state: “When using features from ImageNet classifiers, we set the learning rate to 1*10^-4”. This larger learning rate is used for both cases: G initialized, and G not initialized. Therefore, we use the learning rate 1*10^-4 for training the models that produce our best results.
>
> The difference of 0.05 in Inception Score for the result pointed by the reviewer, 7.67 (ours) vs 7.72 (Warde-Farley & Bengio, 2017) is not statistically significant and should not be used as a demerit for our paper.
> Our results are at the same level as the state-of-the-art GANs and other MMD methods. In fact, our quantitative results (including our models without G initialization) are way better than recent MMD GAN methods (as you can See in Table 3).

---

> ### Author Response · Authors · 2018-11-24
> **Reply to AnonReviewer4  (Part 1 / 3):**
>
>
> We would like to thank the reviewer for the questions and comments.
> In order to better address your concerns, we have uploaded a new version of the paper that contains three new appendices: A.11, A.12 and A.13. Please see below our detailed reply for your questions/comments. We believe that the new added appendices and the clarifications given in our reply will address all the concerns and misconceptions of the reviewer.
>
> 1) About the loss function and novelty:
>
> AUTHORS:  Regarding the objective function, please note that, in the paper, we never mentioned that our loss function is novel, nor that we were the first to perform feature matching. As you mentioned, one of the key novelties is on avoiding the min/max game by using a pretrained feature extractor, which sets our work completely apart from [Salimans et al., 2016] and [Warde-Farley & Bengio, 2017]. In addition, notice that differently from [Salimans et al., 2016] and [Warde-Farley & Bengio, 2017] we perform feature matching using all the layers of the network and this is extremely important for the good performance of the method (as you can see in Table 2).
>
> The research community in generative models has spent the last few years trying to figure out methods to make GANs training more stable. Here we offer an alternative novel method that is competitive with GANs while moving away completely from the problematic min/max game. Hence, it is safe to say that the paper presents a relevant novel contribution for the research community in generative models.
>
> 2)  Muli-scale features (on multiple layers of the CNN)  prevent degeneracy of the generator :
>
> AUTHORS:  Please note that our loss does not perform matching using a single feature map, it uses many layers to do feature matching, and this is crucial to prevent such degeneracy. As we are matching effectively on f_1... f_m where m are are features extracted on different layers, we are getting matching on different scales of the generated image. Hence the claim of the reviewer that nothing prevents degeneracy in the objective is not true, and this is not an artifact of the features, it is due to the multi-scale nature of the features!  This multi-feature matching on multiple scales regularizes the learning of the generator. We quote here [1]: " representations obtained across the layers of a CNN increasingly capture the statistical properties of natural images, producing impressive texture synthesis results".
>
> Note that this multi-features matching has been exploited also in end to end style transfer, or super-resolution [1], but it is novel in generative modeling context . In [1] for instance, the authors show that by matching CNN layers of a pretrained network  one can do super-resolution, we push these observation further by showing that those multi scale features are sufficient statistics also for generation.  Moreover  our empirical results on generation from those deep priors from pretrained features complements theoretical results in [2], that gives theoretical guarantees  for signal recovery from features of deep networks in inverse problems.
>
> [1] Bruna et al. Super-Resolution with Deep Convolutional Sufficient Statistics. ICLR 2016, https://arxiv.org/pdf/1511.05666.pdf
> [2]Global Guarantees for enforcing deep generative priors by empirical risk minimization, Hand et al.
> https://arxiv.org/pdf/1705.07576.pdf

---

### Author Response · Authors · 2018-11-13
**We have submitted a revised version of the paper.**


In order to better address the questions and suggestions from the reviewers, we have uploaded a revised version of the paper. We have done a careful proof reading on the paper, corrected all the typos pointed and greatly improved/expanded some sections and added new ones, as follows:

(1) We have improved and expanded “Sec. 2.4. Matching Features with ADAM Moving Average”, which now includes a more detailed description of our proposed Adam Moving Average (AMA); and also brings more information regarding the motivation and intuition behind AMA;

(2) We have improved and expanded “Sec 4.2.3. Adam Moving Average and Training Stability”, which now contains more detailed experiments to further demonstrate the advantage of AMA over the simple Moving Average (MA). This section also contains more discussions about AMA vs MA experimental results;

(3) We have added a new section “4.3 Discussion”, which contains a more thorough discussion about the experimental results and comparison with state-of-the-art adversarial methods and MMD-based methods.

We have added two new appendices:
(1) “A.9. Impact of Adam Moving Average for VGG19 feature extractor.”, which presents experimental results that also indicate the advantage of AMA over MA when VGG19 feature extractor is employed.

(2) “A.10. Visual Comparison between GFMN and GMMN Generated Images.”, which shows a visual comparison between images generated by GFMN and GMMN (Li et al., 2015).

(3) "A.11. Sampling from the Pretrained Decoder vs. Sampling from GFMN models.",  where we present a comparison between images generated by sampling from the decoder (with no additional training from GFMN) and images generated by GFMN.

(4) "A.12. Impact of using Global Mean Features vs. Minibatch-wise Mean Features of the REAL data.", where we demonstrate that if we have a mean for all the real dataset (global mean), one still needs moving average for the generator (fake) and that adam moving average outperforms the simple Moving average.

(5) "A.13. Autoencoder features vs. VGG19 features for CelebA.", which presents results that evidence the superiority of VGG19 features for CelebA dataset.

*** Appendices A11, A12 and A13 added after we received the review from Rev #4 ***


We hope we addressed here the main concerns of the reviewers and that the new revised paper will help them in further appreciating the technical contributions of the paper and in  improving their overall assessment. We think our paper brings an exciting result to the deep learning community that conveys a simple  and yet exciting message:  feature matching in the space of pre-trained deep CNN allows an efficient training of generative models that circumvents the cumbersome min/max game in GANs.

---

### Public Comment · (anonymous) · 2018-11-20
**Concerning the theoretical guarantees**

It seems the method proposed in the paper is to use the features of a pretrained network to build a kernel for moment matching. Of course, these features being finite dimensional, the associated kernel is not necessarily universal and therefore can't provide any convergence guarantee for the method.
Fortunately, this might be fixable by doing a Kronecker sum of these features and, for example, features of a RBF kernel (or another characteristic kernel). This translate in practice by a simple sum of kernels, which means using something like the MMD GAN loss as a regularizer.

---

> ### Author Response · Authors · 2018-11-21
> **Reply to "Concerning the theoretical guarantees"**
>
>
> Thank you for your interest in our work.
>
> We agree with you that one can use the tensor product or sum of a RBF kernel features and  the pretrained CNN features. However, while this will give some theoretical guarantees as studied in Li et al (2017) [MMD GAN], this will result in an unpractical algorithm since the dimensionality of our feature space is very big (hundreds of thousands of features, which will render the tensor product  even more expensive).
> One can also compose the pretrained features  with RBF kernels as  in Li et al (2017) [MMD GAN]  and Li et al (2015) [GMMN] but this will be also computationally expensive (because of the large feature space of pretrained networks) and will need also large mini batches to estimate the MMD, and we loose all appealing practical advantages of GFMN. The goal of the paper is to show that using pretrained features with a linear kernel solely is enough for training state-of-the-art generators. In other words, we show that the pretrained deep CNN features provide sufficient statistics for image generation.
>
> Moreover, please note that one of the most interesting properties of our approach is that it circumvents the cumbersome min/max game in adversarial methods. If we use the (adversarial) MMD GAN loss as you suggested, we would have to train under the problematic min/max game. Additionally, as you can see in Table 3, our Inception Score for CIFAR10 is significantly better than the Inception Score from Li et al (2017) [MMD GAN], which shows that, in practice, our proposed approach is very effective.

---

### Public Comment · ~Suman_Ravuri1 · 2018-12-03
**Interesting Paper**

This was a thought-provoking paper, and it was pretty surprising that the authors could train a generator using relatively few features (< 1M). One important part of the algorithm seems to be the ADAM moving average, which the authors claim is more stable that the simple moving average (MA). Perhaps more important than the stability is that ADAM moving averages are divided by the square root of the second moment. I think that the scaling is probably what makes AMA work much better than MA. It would be interesting to see if replacing AMA with a simple moving average of means/variances and dividing each feature by the MA of the standard deviation could achieve to same effect.

Also, to clear up a minor misunderstanding -- that "MoLM-1536 can be used in large-scale environments only, while GFMN can be used in single GPU environments." --, all of our MoLM experiments used a single GPU (though that GPU was a Tesla V/P100).

Also, minor typo: pg. 7 "Yhere is a large boost" -> "There is a large boost"

---

> ### Author Response · Authors · 2018-12-03
> **Reply to "Interesting Paper"**
>
>
> Hi Suman Ravuri,
>
> Thank you for the positive comments and suggestions.  We are glad that you think the paper is “thought-provoking”, this was indeed one of the intentions of the work.
>
> Regarding your suggestion of “replacing AMA with a simple moving average of means/variances and dividing each feature by the MA of the standard deviation”, we just tried that, but the results are quite similar to the ones of simple MA.
>
> Note that in Adam moving average the variance terms are not on the features, but they are on the gradient of the least square loss between v_j and the difference of means Delta_j= mean_real-mean_fake (Eq. 4). This gradient is "v_j-Delta_j".  The improvements are mainly due the stability of the optimization that adam offers in the non-stationary setting (Delta_j changes as we update the generator). Also note that our goal is not to get a better estimate of each feature mean (real and fake) alone, we would like to get a better estimate of mean differences (mean_real-mean_fake). This is arguably better, since the fake mean is non stationary and if we have errors from single means estimation this may add up in their difference.
>
> Thank you for the clarification regarding the training of your models. We will adjust our text as soon as we have the opportunity to upload a new version of the paper.  We have trained our models using K80/40 GPUs which contains less memory than V/P100. That’s why we thought your model (which uses significantly more moments) would need additional resources than what we have used.
>
> Thank you for pointing out the typo, it will be corrected in the new version of the text.

---

### Meta-Review · Area_Chair1 · 2018-12-17
**A well-executed piece of empirical work with unfortunately little to offer in the way of novel ideas.**

**Confidence:** 3
**Recommendation:** Reject

**Metareview:**

The paper proposes a method of training implicit generative models based on moment matching in the feature spaces of pre-trained feature extractors, derived from autoencoders or classifiers. The authors also propose a trick for tracking the moving averages by appealing to the Adam optimizer and deriving updates based on the implied loss function of a moving average update.

It was generally agreed that the paper was well written and easy to follow, that empirical results were good, but that the novelty is relatively low. Generative models have been built out of pre-trained classifiers before (e.g. generative plug & play networks), feature matching losses for generator networks have been proposed before (e.g. Salimans et al, 2016).  The contribution here is mainly the  extensive empirical analysis plus the AMA trick.

After receiving exclusively confidence score 3 reviews, I sought the opinion of a 4th reviewer, an expert on GANs and GAN-like generative models. Their remaining sticking points, after a rapid rebuttal, are with possible degeneracies in the loss function and class-level information leakage from pre-trained classifiers, making these results are not properly "unconditional". The authors rebutted this by suggesting that unlike Salimans et al (2016), there is no signal backpropagated from the label layer, but I find this particularly unconvincing: the objective in that work maximizes a "none-of-the-above" class (and thus minimizes *all* classes). The gradient backpropagated to the generator is uninformative about which particular class a sample should imitate, but the features learned by the discriminator needing to discriminate between classes shape those gradients in a particular way all the same, and the result is samples that look like distinct CIFAR classes. In the same way, the gradients used to train GFMN are "shaped" by particular class-discriminative features when trained against a classifier feature extractor.

From my own perspective, while there is no theory presented to support why this method is a good idea (why matching arbitrary features unconnected with the generative objective should lead to good results), the idea of optimizing a moment matching objective in classifier feature space is rather obvious, and it is unsurprising that with enough "elbow grease" it can be made to work. The Adam moving average trick is interesting but a deeper analysis and ablation of why this works would have helped convince the reader that it is principled.

This paper was very much on the borderline. Aside from quibbles over the fairness of comparisons above, I was forced to ask myself whether I could imagine that this would be a widely read, influential, and frequently cited piece of work. I believe that the carefully done empirical investigation has its merits, but that the core ideas are rather obvious and the added novelty of a poorly understood stabilized moving average is not enough to warrant acceptance.